# Airplane Vortices Evolution Near Ground

**Josep M. Duró** 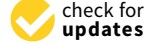 **and Josep M. Bergadà \***

Department of Fluid Mechanics, Universitat Politècnica de Catalunya, 08034 Barcelona, Spain;
jm.duro21@gmail.com
**\*** Correspondence: josep.m.bergada@upc.edu; Tel.: +34-937-398-771

**Abstract:** Airport traffic around the world has sharply increased over the years; as a result, airports need to be enlarged and the landing or taking off times between two consecutive airplanes must be reduced. To precisely determine the minimum time required between two consecutive airplanes, it is essential to understand the main physical characteristics of the vortices generated under airplanes' wings and their evolution under different atmospheric conditions. In the present paper, such information is obtained through the complex potential equation of a vortex together with the potential Bernoulli equation. The process starts with the characteristic complex potential equation, which is simplified to find the velocity potential function. Then, the temporal movement of the vortices' central core, the velocity and pressure fields around the vortical structures and the effect of the crosswind on the vortices' displacement, velocity and pressure fields are obtained. The paper shows how optimizing the process of measuring and calculating the vortices' behavior could save a certain amount of time between airplanes, therefore increasing airport throughput. This paper introduces a potential flow method, which is coupled with the temporal variation of the flow circulation, to predict the vortices' behavior and movement over time. The inclusion of circulation decay over time is employed to simulate the viscosity effect over the vortical structures. The in-house code generates results in less than one minute and needs to be seen as a tool to determine, for each airport and crosswind condition, the minimum time needed between two consecutive airplanes.

**Keywords:** under wing vortex; airport throughput optimization; potential flow

## 1. Introduction

The study of the vortex wake evolution generated by an airplane in ground proximity is a very interesting field in airport management, as it could be used in order to optimize the time between two consecutive airplanes during take-off or landing, hence increasing the airport traffic and throughput. The time between two consecutive airplanes needs to be large enough to avoid wake interaction between the previous airplane and the following one; otherwise the risk of having a crash rises.

It is particularly interesting to employ potential flow theory to evaluate vortices' evolution; then the computational time required to perform the calculations is very low and therefore the calculations could be done in situ, which may be convenient for safety requirements. For example, if the simulation is desired to be done using Computational Fluid Dynamics (CFD), the computational time would be extremely high due to the large mesh needed to cover the extensive domain. In any case it needs to be kept in mind that the main problem with the use of potential flow theory is that it does not take into account the effect of fluid viscosity, and thus there is no dissipation of the vortex and its temporary behavior may not be fully precise. In the present paper this problem is minimized thanks to the use of time-dependent circulation.

Nowadays, the situation is the following. The regulation of the time between airplanes is established by the International Civil Aviation Organization (ICAO) and divides the different airplanes into four categories depending on their weight (Campos et al. [1]). The time between two consecutive airplanes needs to be large enough to secure that there



is no possible effect of the previous airplane vortex wake on the following one. The calculation of this time is based on empirical data obtained from a wide range of historical measurements. This categorization has a big safety factor, so it appears that a better knowledge of the vortical structures and their evolution would lead to a time reduction. Another weakness of the method used nowadays is the uncertainties when a new airplane model is incorporated, since there are no empirical measurements available. For example, due to the appearance of big airplanes (such as the A380-800) the categorization rules had to be modified. However, to optimize the procedures a solid and fast method to calculate the characteristics of the vortices is needed, as they can vary either as a function of the atmospheric pressure or the crosswind, among other parameters.

To better understand the actual situation of the research it is interesting to consider the work done by [2], which focuses on the dynamic behavior of the vortical structures produced by airplanes during touchdown, providing valuable information regarding the different instabilities and the vortex decay process. As will be seen later, the understanding of this process is used, in some studies (Wakim et al. [3] and Holzäpfel et al. [4]), to reduce the mentioned instabilities and force a faster propagation of the vortices.

The main method being used in several references [3,5–8] is the measurement of the vortex wake's characteristics through a Doppler LIDAR system, as explained in Smalikho et al. [9]. There are some studies (see Smalikho et al. [6]) where this method is used taking into account the ground effect, with the aim of increasing the precision of the results. Nevertheless, regardless of the method employed, the results are subjected to the intrinsic errors of the Doppler LIDAR radar measurements. The implementation of this experimentally-based methodology requires some simplifications to reduce the number of variables as well as the computational time. The different errors generated at each step of the process build up, providing a bigger final error, even though it is usually considered as acceptable.

On the other hand, some methods have been developed in order to reduce the strength of the vortex wake, for example, in [3] it is stated that the vortex rebound near the runway is produced by the apparition of secondary vortices induced by the boundary layer near the ground. Therefore, the objective of the study was to find a boundary condition that could reduce this phenomenon, concluding that a possibility could be to suck the vortices' boundary layer near the ground. This would avoid the apparition of the secondary vortexes, so the principal ones would have a similar behavior to the inviscid case, whose displacement is hyperbolic, meaning that they move away faster from the runway domain. Another methodology is proposed by Holzäpfel et al. [4]. Their idea is to reduce the strength of the vortex by dividing the main ones into many small vortices, reducing their strength and therefore encouraging the vanishing of the vortex wake.

A long set of measurements to evaluate the wake vortex behavior in ground proximity were performed by Holzäpfel and Steen [10]. They observed that the importance of turbulence and crosswind for wake vortex decay was weak, but light crosswind was found to be sufficient to cause pronounced asymmetric rebound. They concluded that wake rebound predictions could be significantly improved when considering the strength and time of generation of secondary vortices dependent on the crosswind's value. In a further paper [11], the impact of the meteorological parameters on vortices' evolution was experimentally investigated. A ranking of impact parameters affecting such evolution was presented, and it was observed that flow stratification was very important at cruise altitudes.

The suitability of combining several independent wake vortex models to improve deterministic and probabilistic wake vortex forecast was analyzed in Körner et al. [12]. In Smalikho et al. [13], they determined for different types of aircraft and wind turbulence strengths, the limits of applicability of the radial velocity method for estimating wake vortex parameters.

Recently there has been an increase in the use of CFD methods, which can give a very precise understanding of the phenomenon. However, there are two main problems associated with this methodology as stated in [14]. First, the simulations should be done

in 3D, which involves large computational resources and time, and the results vary depending on the chosen turbulence model. Secondly, the simulations depend as well on the environmental conditions, meaning that to cover the whole range of possible cases it would be necessary to perform many 3D simulations. Although the second problem is understandable due to the characteristics of each case, the first problem is the main one, especially when the objective is to obtain fast results. In other CFD simulations Lin et al. [15], researchers tried to couple the wake's roll-up process with its own vortices to initialize the wake. But again, the main problem is the high computational time and computer resources required to undertake such simulations.

One of the objectives of the present article is to develop a program based on potential flow theory that can produce accurate results (even though with a reasonable error) and needs less computational time when compared to existing CFD methods. The paper is outlined as follows: the mathematical process will start from the known complex potential equation of a vortex and its image. Then, by simplifying it and separating the real and the imaginary parts, the velocity potential equation can be obtained. Afterwards, from this equation, the temporal variation and the velocity field can be found. Finally, with the Bernoulli equation for potential flow, the pressure distribution across the entire domain will be obtained. The validation of the model and the results section are presented next, and the paper ends with the conclusions.

## 2. Mathematical Procedure to Obtain the Velocity and Pressure Fields

Based on the potential flow theory, the velocity potential equation to find the velocity field in the desired domain can be obtained. Once the temporal variation of the potential is known, the pressure field can be found through the Bernoulli equation for potential flow, which reads:

$$\rho\left(\frac{\partial\phi}{\partial t} + \frac{V^2}{2}\right) + P = P_0 \tag{1}$$

The procedure starts considering two real vortices situated at points $Z_0 = (X_0, Y_0)$ and $Z_1 = (-X_0, Y_0)$, see Figure 1. In order to simulate the presence of the ground, two mirror vortices located in the positions $Z_0^* = (X_0, -Y_0)$ and $Z_1^* = (-X_0, -Y_0)$ have to be considered.

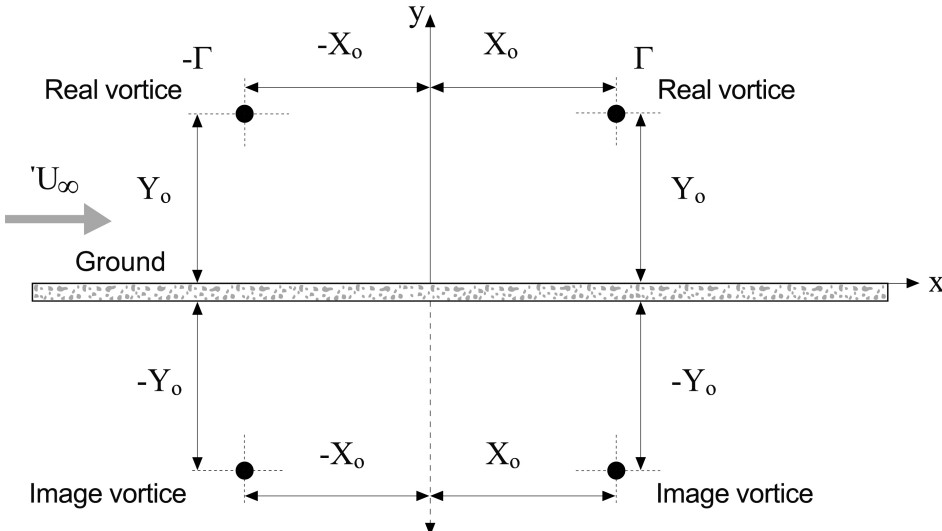

**Figure 1.** Coordinate system and vortices position.

The complex potential function of two vortices, one real and its image, is defined using Equation (2); this equation also considers the effect of the free stream velocity.

The properties of such functions are defined in [16]; based on this information, the final form of the complex potential function has to be the one presented in Equation (3).

$$f(Z) = U_\infty Z + \frac{i\Gamma}{2\pi} ln\left(\frac{Z - Z_0^*}{Z - Z_0}\right) \tag{2}$$

$$f(Z) = \phi + i\psi \tag{3}$$

After substituting in Equation (2) the position of the vortices located on the right-hand side of the domain and rearranging the real and imaginary parts, we obtained:

$$f(Z) = U_\infty(X + iY) + \frac{i\Gamma}{2\pi} ln\left(\frac{(X - X_0) + i(Y + Y_0)}{(X - X_0) + i(Y - Y_0)}\right) \tag{4}$$

Recalling the concept of a logarithm of a complex function, Equation (4) can be expressed as:

$$\begin{aligned} f(Z) = U_\infty(X + iY) &+ \frac{i\Gamma}{2\pi}\left[ln|(X - X_0) + i(Y + Y_0)| + i \quad \text{atan}\left(\frac{Y + Y_0}{X - X_0}\right)\right] \\ &- \frac{i\Gamma}{2\pi}\left[ln|(X - X_0) + i(Y - Y_0)| + i \quad \text{atan}\left(\frac{Y - Y_0}{X - X_0}\right)\right] \end{aligned} \tag{5}$$

Once the logarithmic terms from Equation (5) have been multiplied by their conjugate and the real and imaginary terms separated, the resulting velocity potential and stream functions characteristic for the vortex situated at the top right zone and due to the influence of the right part of the domain, will take the form:

$$\phi_r(x, y) = U_\infty X + \frac{\Gamma}{2\pi}\left[-\text{atan}\left(\frac{Y + Y_0}{X - X_0}\right) + \text{atan}\left(\frac{Y - Y_0}{X - X_0}\right)\right] \tag{6}$$

$$\psi_r(x, y) = U_\infty Y + \frac{\Gamma}{4\pi}\left\{ln\left[(X - X_0)^2 + (Y + Y_0)^2\right] - ln\left[(X - X_0)^2 + (Y - Y_0)^2\right]\right\} \tag{7}$$

Taking into account that Equations (6) and (7) only consider the effect of the two right-hand side vortices, the real and the imaginary one, it is necessary to add the influence from the other two remaining vortices depicted in Figure 1. It needs to be remembered that the effect of free stream velocity needs to be considered only once; therefore when applying Equation (2) to the other two vortices, the free stream velocity does not need to be taken into account, and then the free stream velocity would be evaluated twice (It is important to note that the potential theory is linear). Following the procedure just defined, the influence due to the left part of the domain can be represented as:

$$\phi_l(x, y) = \frac{\Gamma}{2\pi}\left[\text{atan}\left(\frac{Y + Y_0}{X + X_0}\right) - \text{atan}\left(\frac{Y - Y_0}{X + X_0}\right)\right] \tag{8}$$

$$\psi_l(x, y) = -\frac{\Gamma}{4\pi}\left\{ln\left[(X + X_0)^2 + (Y + Y_0)^2\right] - ln\left[(X + X_0)^2 + (Y - Y_0)^2\right]\right\} \tag{9}$$

After performing the addition of the two corresponding functions, the resulting potential and stream functions are represented as:

$$\phi(x, y) = \phi_r(x, y) + \phi_l(x, y) \tag{10}$$

$$\begin{aligned} \phi(x, y) = U_\infty X + \frac{\Gamma}{2\pi}\Bigg[&-\text{atan}\left(\frac{Y + Y_0}{X - X_0}\right) + \text{atan}\left(\frac{Y - Y_0}{X - X_0}\right) + \\ &+\text{atan}\left(\frac{Y + Y_0}{X + X_0}\right) - \text{atan}\left(\frac{Y - Y_0}{X + X_0}\right)\Bigg] \end{aligned} \tag{11}$$

$$\psi(x, y) = \psi_r(x, y) + \psi_l(x, y) \tag{12}$$

$$\psi(x,y) = U_\infty Y + \frac{\Gamma}{4\pi}\Big(ln\big[(X-X_0)^2 + (Y+Y_0)^2\big] - ln\big[(X-X_0)^2 + (Y-Y_0)^2\big]$$
$$-ln\big[(X+X_0)^2 + (Y+Y_0)^2\big] + ln\big[(X+X_0)^2 + (Y-Y_0)^2\big]\Big) \tag{13}$$

From Equation (13) and through the CauchY–Riemann relations (Equations (14) and (15)) the fluid velocity can be found across the entire domain:

$$V_X = \frac{\partial\psi}{\partial y} = U_\infty + \frac{\Gamma}{2\pi}\Bigg[\frac{Y+Y_0}{(X-X_0)^2 + (Y+Y_0)^2} - \frac{Y-Y_0}{(X-X_0)^2 + (Y-Y_0)^2} +$$
$$-\frac{Y+Y_0}{(X+X_0)^2 + (Y+Y_0)^2} + \frac{Y-Y_0}{(X+X_0)^2 + (Y-Y_0)^2}\Bigg] \tag{14}$$

$$V_Y = -\frac{\partial\psi}{\partial x} = -\frac{\Gamma}{2\pi}\Bigg[\frac{X-X_0}{(X-X_0)^2 + (Y+Y_0)^2} - \frac{X-X_0}{(X-X_0)^2 + (Y-Y_0)^2} +$$
$$-\frac{X+X_0}{(X+X_0)^2 + (Y+Y_0)^2} + \frac{X+X_0}{(X+X_0)^2 + (Y-Y_0)^2}\Bigg] \tag{15}$$

Finally, the last unknown parameter from Equation (1) is the temporal variation of the velocity potential, and it is found by means of the chain rule:

$$\frac{\partial\phi}{\partial t} = \frac{\partial\phi}{\partial X_0}\frac{\partial X_0}{\partial t} + \frac{\partial\phi}{\partial Y_0}\frac{\partial Y_0}{\partial t} = V_{X0}\frac{\partial\phi}{\partial X_0} + V_{Y0}\frac{\partial\phi}{\partial Y_0} \tag{16}$$

The terms $\frac{\partial X_0}{\partial t}$ and $\frac{\partial Y_0}{\partial t}$ represent the velocity of the right-hand side real vortex. To determine these terms, it is necessary to consider the effect of the other three vortices on it. To do so it is necessary to calculate the reduced potential function, which takes the following form:

$$f^*(z) = \phi^* + i\psi^* = U_\infty Z + \frac{i\Gamma}{2\pi}ln(Z - Z_0^*) - \frac{i\Gamma}{2\pi}ln\left(\frac{Z - Z_1^*}{Z - Z_1}\right) \tag{17}$$

The different positions of the vortices cores are:

$$Z_0^* = (X_0 - iY_0); Z_1 = (-X_0 + iY_0); Z_1^* = (-X_0 - iY_0); Z = (X + iY) \tag{18}$$

Substituting Equation (18) in Equation (17) and after rearranging the real and imaginary terms, we obtain:

$$\phi^* + i\psi^* = U_\infty X + iU_\infty Y + \frac{i\Gamma}{2\pi}ln((X-X_0) + i(Y+Y_0))$$
$$-\frac{i\Gamma}{2\pi}ln((X+X_0) + i(Y+Y_0)) + \frac{i\Gamma}{2\pi}ln((X+X_0) + i(Y-Y_0)) \tag{19}$$

Remembering the concept of the logarithm of a complex function, once the logarithmic terms from Equation (19) have been multiplied by their conjugate and the real and imaginary terms separated, the resulting reduced potential and stream functions take the form:

$$\phi^* = U_\infty X - \frac{\Gamma}{2\pi}\left[atan\frac{Y+Y_0}{X-X_0}\right] + \frac{\Gamma}{2\pi}\left[atan\frac{Y+Y_0}{X+X_0}\right] + \frac{\Gamma}{2\pi}\left[atan\frac{Y-Y_0}{X+X_0}\right] \tag{20}$$

$$\psi^* = U_\infty Y + \frac{\Gamma}{4\pi} ln((X - X_0)^2 + (Y + Y_0)^2) - \frac{\Gamma}{4\pi} ln((X + X_0)^2 + (Y + Y_0)^2)$$
$$+ \frac{\Gamma}{4\pi} ln((X + X_0)^2 + (Y - Y_0)^2)$$
(21)

The velocity of the vortex located at the initial point $(X_0; Y_0)$ is defined as:

$$V_{X0} = \frac{\partial X_0}{\partial t} = \left.\frac{\partial \phi^*}{\partial X}\right|_{X=X_0;\ Y=Y_0} = \left.\frac{\partial \psi^*}{\partial Y}\right|_{X=X_0;\ Y=Y_0}$$
(22)

$$V_{Y0} = \frac{\partial Y_0}{\partial t} = \left.\frac{\partial \phi^*}{\partial Y}\right|_{X=X_0;\ Y=Y_0} = -\left.\frac{\partial \psi^*}{\partial X}\right|_{X=X_0;\ Y=Y_0}$$
(23)

Then the resulting equations take the form:

$$V_{X0} = V_X(X_0, Y_0) = U_\infty + \frac{\Gamma}{4\pi}\left[\frac{1}{Y_0} - \frac{Y_0}{(X_0^2 + Y_0^2)}\right] = U_\infty + \frac{\Gamma}{4\pi}\left[\frac{X_0^2}{Y_0(X_0^2 + Y_0^2)}\right]$$
(24)

$$V_{Y0} = V_Y(X_0, Y_0) = \frac{\Gamma}{4\pi}\left[\frac{X_0}{(X_0^2 + Y_0^2)} - \frac{1}{X_0}\right] = -\frac{\Gamma}{4\pi}\left[\frac{Y_0^2}{X_0(X_0^2 + Y_0^2)}\right]$$
(25)

To obtain the terms $\dfrac{\partial \phi}{\partial X_0}$ and $\dfrac{\partial \phi}{\partial Y_0}$ it will be necessary to make the derivative of Equation (11), this is straightforward and the outcome is presented below.

$$\frac{\partial \phi}{\partial X_0} = \frac{\Gamma}{2\pi}\left[-\frac{Y + Y_0}{(X - X_0)^2 + (Y + Y_0)^2} + \frac{Y - Y_0}{(X - X_0)^2 + (Y - Y_0)^2} + \right.$$
$$\left. -\frac{Y + Y_0}{(X + X_0)^2 + (Y + Y_0)^2} + \frac{Y - Y_0}{(X + X_0)^2 + (Y - Y_0)^2}\right]$$
(26)

$$\frac{\partial \phi}{\partial Y_0} = \frac{\Gamma}{2\pi}\left[-\frac{X - X_0}{(X - X_0)^2 + (Y + Y_0)^2} - \frac{X - X_0}{(X - X_0)^2 + (Y - Y_0)^2} + \right.$$
$$\left. +\frac{X + X_0}{(X + X_0)^2 + (Y + Y_0)^2} + \frac{X + X_0}{(X + X_0)^2 + (Y - Y_0)^2}\right]$$
(27)

Then the equation giving the temporal variation of the velocity potential at any point of the domain takes the form:

$$\frac{\partial \phi}{\partial t} = \frac{\Gamma}{2\pi}\left(-\frac{Y+Y_0}{(X-X_0)^2+(Y+Y_0)^2} + \frac{Y-Y_0}{(X-X_0)^2+(Y-Y_0)^2} + \right.$$
$$\left. -\frac{Y+Y_0}{(X+X_0)^2+(Y+Y_0)^2} + \frac{Y-Y_0}{(X+X_0)^2+(Y-Y_0)^2}\right)\left(U_\infty + \frac{\Gamma}{4\pi}\left[\frac{X_0^2}{Y_0(X_0^2+Y_0^2)}\right]\right) +$$
$$+\frac{\Gamma}{2\pi}\left(-\frac{X-X_0}{(X-X_0)^2+(Y+Y_0)^2} - \frac{X-X_0}{(X-X_0)^2+(Y-Y_0)^2} + \right.$$
$$\left. +\frac{X+X_0}{(X+X_0)^2+(Y+Y_0)^2} + \frac{X+X_0}{(X+X_0)^2+(Y-Y_0)^2}\right)\left(-\frac{\Gamma}{4\pi}\left[\frac{Y_0^2}{X_0(X_0^2+Y_0^2)}\right]\right)$$
(28)

Finally, substituting Equations (14), (15) and (28) into the potential Bernoulli equation (Equation (1)), the relative pressure along the flow domain can be found.

## 3. Validation

Before the validation, it is necessary to take into account another feature that will also be introduced to the study. This is the temporal variation of the vortex's circulation. Then due to the viscosity effect, circulation cannot be considered constant. In fact, the time-

dependent vortex circulation is expected to have considerable effects on the vortices' temporal behavior, for example their lateral motion will be slowed down because the circulation decrease with time. According to [6], the simplest expression that models the temporal evolution of the vortex's circulation is the single-parameter wake vortex decay, represented as Equation (29):

$$\Gamma(t) = \Gamma_0 \cdot exp\left(-\frac{t}{T_{e^{-1}}}\right) \tag{29}$$

For solving Equation (29), the parameter $T_{e^{-1}}$ must be known, which is retrieved from experimental data. Gerz et al. [17] stated that the dependence of the decay can be related to a time parameter $t_0$, whose value is obtained for each case imposed by the initial circulation. Based on references [6,17], the mentioned value will be defined as $T_{e^{-1}} = 10t_0/\pi$, where $t_0$ is given as:

$$t_0 = \frac{2\pi}{\Gamma_0}\left(\frac{\pi}{4}B\right)^2 \tag{30}$$

The $\Gamma_0$ and $B$ values are the vortex's initial circulation and the wingspan of the airplane, respectively. Instead of using this temporal variation, another relation can be used, which provides a two-phase decay of the vortices' circulation [10]. This second temporal variation is more complex and realistic when there is low ambient turbulence, but some probabilistic data must be used. In this paper, the temporal equation presented in Smalikho et al. [6] is initially employed.

Once Equation (29) is coupled to the study, by means of substituting the value retrieved from this equation at each time step to Equations (1), (14), (15) and (28), the initial validation of the mathematical procedure shall be done. To validate the in-house program with the one-phase decay theory (Equation (29)), since the temporal variation of the circulation simply depends on the initial circulation, the main focus is put on variables whose mathematical formulation is different in each method, for example the velocity domain and the position of the vortices' center throughout time.

The comparison between the results obtained in the present paper and the ones presented by Smalikho et al. [6] are introduced in Figure 2. The vortex central core's vertical and horizontal velocity distribution throughout time, the vortex's vertical position and the separation between the two vortices as a function of time are represented. The temporal variation of the vortex's central position is obtained by combining Equations (24) and (25) together with the time elapsed, and the separation between vortices is drawn from the horizontal velocity and the time elapsed. The evolution of the variables is almost the same in all figures; minor differences are observed in the vortices' positions and separation between vortices. Therefore, the results obtained from the present equations and methodology need to be seen as accurate.

The comparison was done using the initial values introduced in Table 1 and extracted from reference [6], where $\rho$ stands for the air density, the height is the vertical distance from the ground at which the vortex is formed, $\Gamma_0$ is the initial value of the circulation and $B$ is the wing span of the airplane.

**Table 1.** Initial values used for the validation between one-phase decay methods.

| |
|---|
| $\rho = 1.225 \text{ kg/m}^3$ |
| Height = 50 m |
| $\Gamma_0 = 500 \text{ m}^2/\text{s}$ |
| $B = 63.7$ m |

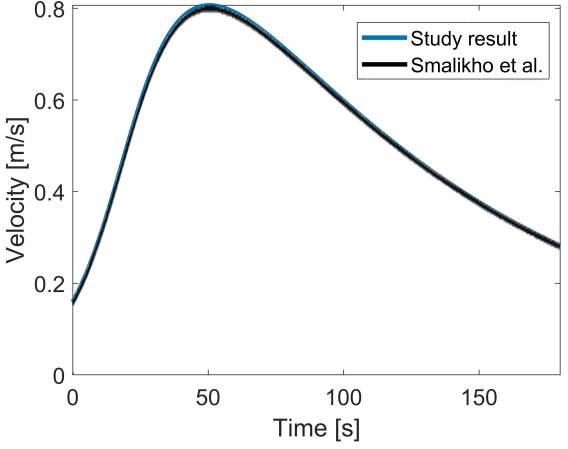

(**a**) Horizontal velocity of the vortex center.

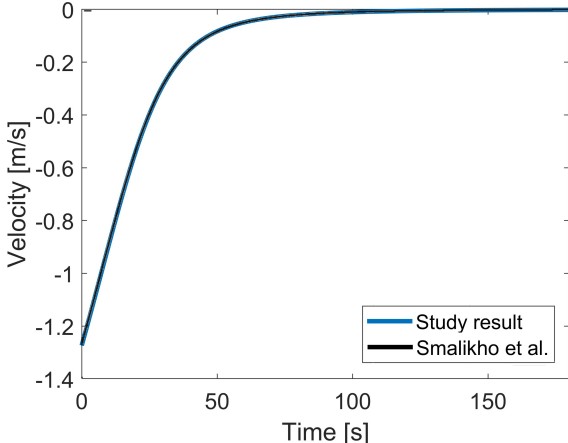

(**b**) Vertical velocity of the vortex center.

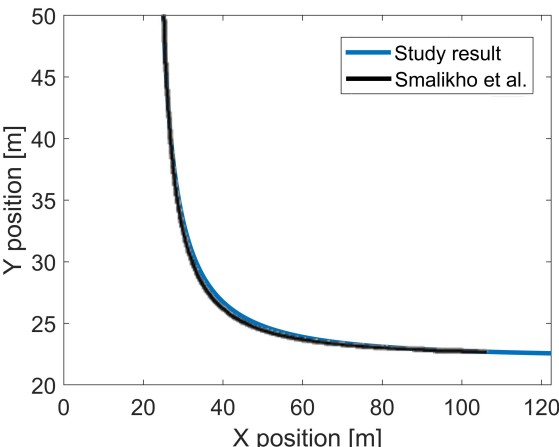

(**c**) Temporal variation of the vortex center position.

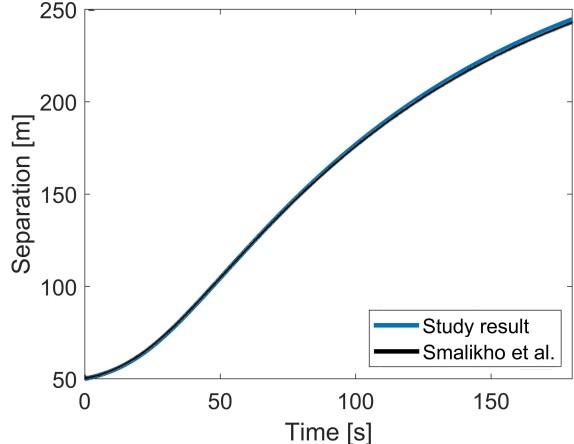

(**d**) Separation between vortex centers throughout time.

**Figure 2.** Validation of the in-house program's mathematical model. Comparison between the present study's results and the ones from Smalikho et al. [6]. Equation (29) has been used to determine the temporal variation of the circulation. Then, to validate the study, the temporal variation of the vortices' position throughout time was plotted.

Once it is seen that the mathematical model is accurate regarding other theories with one-phase decay, the next step is to perform a validation comparing to a two-phase decay model (obtained from reference [10]). The comparison was done using the initial values introduced in Table 2 and extracted from the same reference. The meaning of the parameters presented in Table 2 are the same as the ones initially introduced in Table 1.

The experimentally obtained circulation temporal variation from the mentioned reference has been parameterized, by means of a 6th grade polynomial regression, to include it into the in-house program. The concept is to check how accurate are the results when the circulation temporal decay obtained experimentally is employed. The temporal variation of the circulation for this case is modeled by Equation (32). The results are represented in Figure 3, together with the ones from reference [10], and the ones obtained with Equation (29). To model the temporal circulation, nondimensional values are needed, whose criteria, according to reference [10], are those established in Equation (31). The parameter $b_0$ is as well introduced and expresses the initial separation between vortices.

$$x^* = \frac{x}{b_0}; y^* = \frac{y}{b_0}; \Gamma^* = \frac{\Gamma}{\Gamma_0}; t^* = \frac{t}{t_0}; u^* = \frac{V_{X0}}{w_0}; v^* = \frac{V_{Y0}}{w_0}; w_0 = \frac{\Gamma_0}{2\pi b_0}; b_0 = \frac{\pi}{4}B; \quad (31)$$

**Table 2.** Initial values used for the validation between the in-house program (by using either Equations (29) and (32) for the temporal variation of circulation), and the two-phase decay method presented by Holzäpfel and Steen [10].

| B747-400 |
| --- |
| $\Gamma_0 = 633.3 \ \mathrm{m}^2/\mathrm{s}$ |
| $B = 64.4 \ \mathrm{m}$ |
| $\rho = 1.225 \ \mathrm{kg/m}^3$ |
| Height = 55 m |

$$\Gamma^*_{Val} = 5.71 \cdot 10^{-5} t^{*6} - 1.37 \cdot 10^{-3} t^{*5} + 1.19 \cdot 10^{-2} t^{*4} - 4.12 \cdot 10^{-2} t^{*3} + 3.30 \cdot 10^{-2} t^{*2} - 7.86 \cdot 10^{-2} t^* + 1 \qquad (32)$$

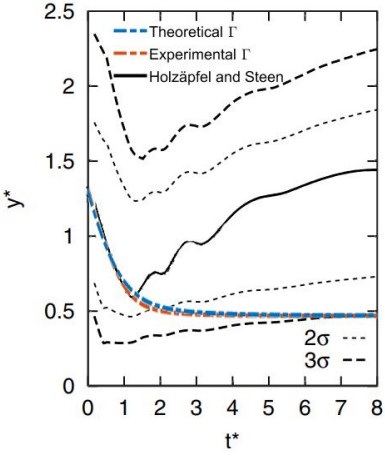

(**a**) Nondimensional vertical position of the vortex throughout time.

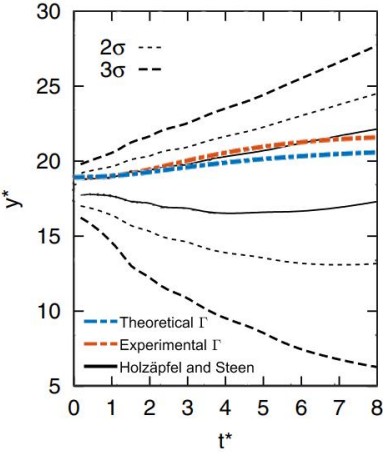

(**b**) Nondimensional horizontal position of the vortex throughout time.

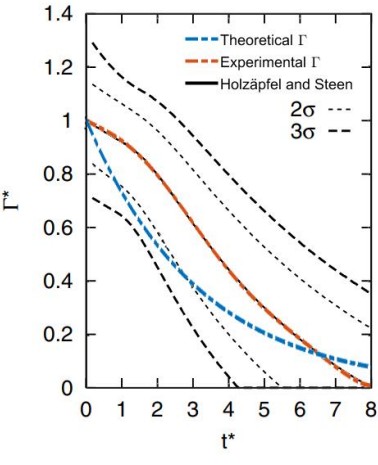

(**c**) Nondimensional circulation of the vortex throughout time.

**Figure 3.** Validation of the in-house program's mathematical model. Comparison between the present study's results and the ones from Holzäpfel and Steen [10]. Figures have been extracted from this reference and edited to be able to later include the in-house program results and have a clearer comparison. However, the probabilistic envelopes of $2\sigma$ and $3\sigma$ have been also included, to see the in-house program results' precision. The results obtained with Equation (29) state for "Theoretical $\Gamma$". The ones obtained with Equation (32) state for "Experimental $\Gamma$". For this case, crosswind is considered null.

The comparison between the results obtained from the in-house program presented in this paper and the ones obtained by Holzäpfel and Steen [10] is shown in Figure 3. From the figures of the vortex center position, it is seen that the deviation of the results obtained by the in-house program is negligible at early times, but as the time increases, the deviation increases. The major discrepancies are observed in Figure 3a, because both curves, the one based on experimental circulation decay and the one using Equation (32), abandon the $3\sigma$ envelope for advanced times. This shows that the accuracy of the results could be better for this case, since the in-house program does not simulate the vortices' rebound. The accuracy of the results is particularly good in Figure 3b; the ones obtained with Equation (32) show an almost perfect accuracy compared to the experimental results for all the time steps. On the other hand, for Equation (29), at early time steps the accuracy is good, but after $t^* = 2$ the prediction becomes loose, even though it does not stray too far from the experimental results. Finally, for Figure 3c, the polynomial regression used for Equation (32) obviously fits perfectly to the experimental results, but the single-parameter temporal variation of $\Gamma$ shows poor accuracy, meaning that it does not fit the experimental temporal variation at any point; additionally, it leaves the $2\sigma$ envelope during $t^* = 1$ to $t^* \approx 2.5$.

The main conclusion from Figure 3 is that the accuracy of the results is better when using experimental circulation decays than when using the circulation decay from Equation (29). Therefore, in the remaining part of the study, experimental circulations will be used to generate the graphs. In the following comparison case, the focus again is being put on the position of the vortices' center. The figures generated to perform the comparison characterize the nondimensional horizontal and vertical vortex position variation throughout time. A polynomial regression of grade 6 has been performed to obtain the circulation temporary decay based on experimental data. The data used for the validation were extracted from reference [11], and involve two cases without crosswind, where the vortices are generated at two different heights. The nondimensionalization was done with the criteria established in Equation (31), according to Holzäpfel [11]. Additionally, the comparison was performed using the initial values introduced in Table 3 and extracted from the same reference.

**Table 3.** Initial values used for the validation of the experimental circulation's temporal variation, comparing with experimental values of the two-phase wake vortex decay theory of Holzäpfel [11].

| A340-300 |
| :---: |
| $\Gamma_0 = 458 \text{ m}^2/\text{s}$ |
| $B = 60.3 \text{ m}$ |
| $b_0 = 47.4 \text{ m}$ |
| $\rho = 1.17 \text{ kg}/\text{ m}^3$ |
| $U_\infty = 0 \text{ m/s}$ |
| $y^* = b_0,\, y^* = 2b_0$ |

With these initial conditions, the temporal variation of $\Gamma$ is given by the following formulas, obtained by performing a polynomial regression of the 6th grade of the experimental circulations obtained in [11]. Equation (33) corresponds to the case with an initial nondimensional height of $b_0$ and Equation (34) corresponds to the case with an initial nondimensional height of $2b_0$.

$$\Gamma^*_{z_0^*=b_0} = 4.86 \cdot 10^{-6} t^{*6} + 6.78 \cdot 10^{-5} t^{*5} - 3.21 \cdot 10^{-3} t^{*4} + 3.26 \cdot 10^{-2} t^{*3} - 0.13 t^{*2} + 9.64 \cdot 10^{-3} t^* + 1 \quad (33)$$

$$\Gamma^*_{z_0^*=2b_0} = 8.61 \cdot 10^{-5} t^{*6} - 2.20 \cdot 10^{-3} t^{*5} + 2.11 \cdot 10^{-2} t^{*4} - 8.96 \cdot 10^{-2} t^{*3} + 0.15 t^{*2} - 0.16 t^* + 1 \quad (34)$$

The comparison between the results obtained from the present paper and the ones presented in Holzäpfel [11] is introduced in Figure 4. Looking at the figures of the vortex's center position, the agreement is very good when considering the horizontal temporal position, especially for higher initial altitudes. When considering the vertical position, the deviation of the results obtained by the in-house program is low at early times, but as the time increases, the deviation also increases. This behavior is given by the fact that as the principal vortices come close to the ground, the vortex structure is more complex, and thus the temporal variation estimated by the potential flow theory is less precise. The vortex behavior could be better modeled if the secondary vortices, which appear due to boundary layer detachment near the ground, were also included in the in-house code. In fact, in page 223 of Holzäpfel and Steen [10], there is a possible parameterization to include those vortices. Some of the data needed to model them are exposed in the mentioned reference. Where, the generation height of the secondary vortexes and its relation with the height of the main ones, and the relation between their circulation and that of the main ones, are given. This aspect could greatly improve the temporary results of the in-house program, as the vortex rebound effect could be simulated.

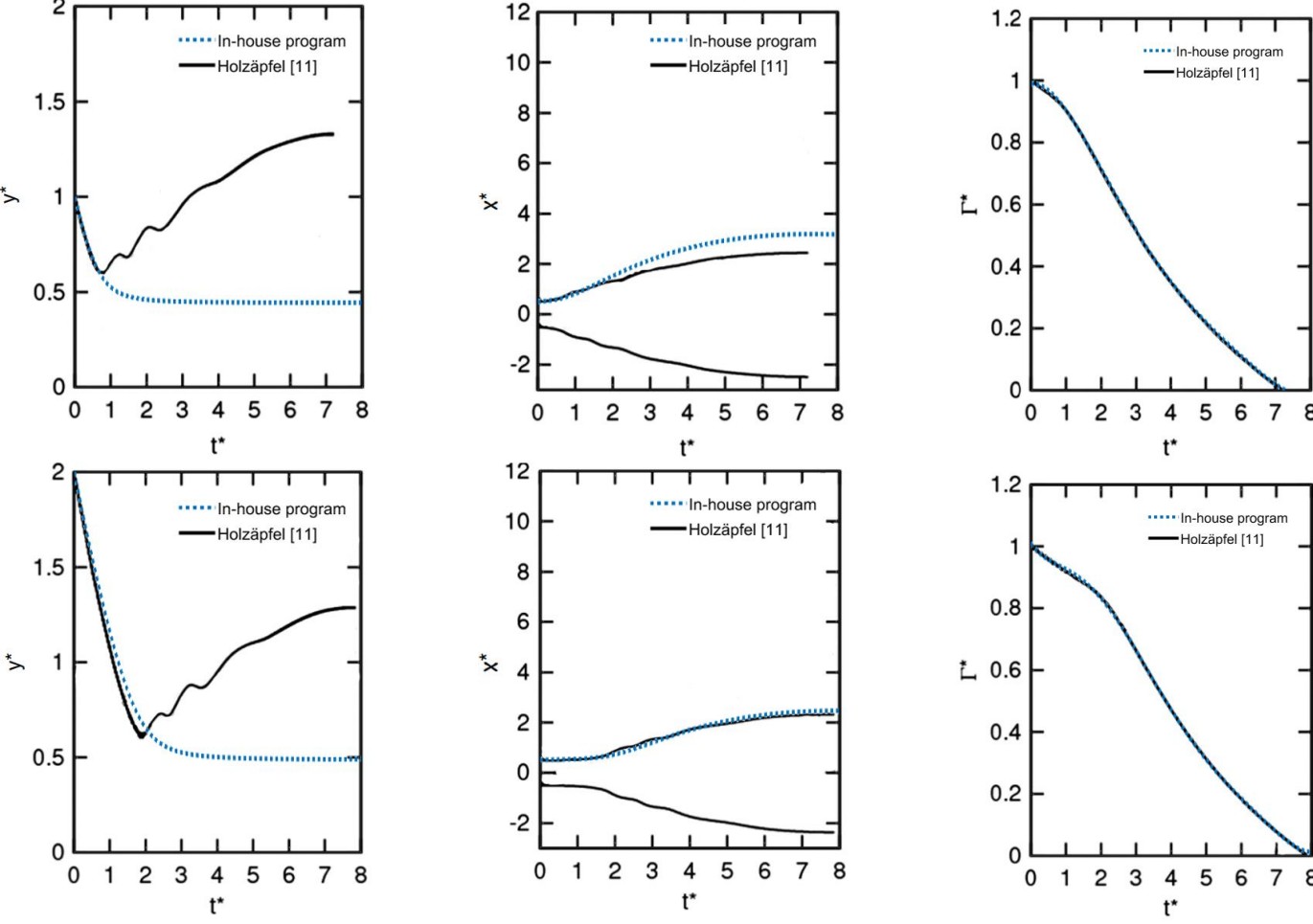

(**a**) Vertical position of the vortex center throughout time.

(**b**) Horizontal position of the vortex center throughout time.

(**c**) Variation of circulation throughout time.

**Figure 4.** Validation of the mathematical model. Comparison between the present study results and the experimental results obtained from [11]. At the upper part of each subfigure the results have been calculated at an initial height equal to $y = b_0$, for the ones at the bottom it is equal to $y = 2b_0$. The in-house program's temporal variation of the circulation has been adjusted to the experimental data retrieved from the mentioned reference. To produce this figure, the results belonging to Holzäpfel [11] have been extracted from the article, and then modified to overlay the results given by the in-house program prediction. No crosswind has been supposed.

Overall, it can be concluded that as the height decreases, the temporal variation of the mathematical model is less precise. In particular, the y* figure of the in-house program has a correct behavior, in both cases, until the vortex rebound phenomena takes place. On the other hand, the x* figure is almost identical, compared to the experimental data, for the altitude $y = 2b_0$, but for an initial height equal to $y = b_0$, the results are only identical until $t^* = 2$, and small deviations are observed after this time. The perturbation due to the ground mostly affects the vertical displacement of the vortex.

## 4. Results and Discussion

In this section we present how the wake vortices evolve for different flow circulations, airplane dimensions and crosswind velocities. In order to have a reference, the ICAO wake turbulence categorization was used [18], where four classes of airplanes depending on their weight are defined; see Table 4 (MTOW stands for Maximum Take-Off Weight). The relation between the MTOW associated with each airplane and the circulation generated was

retrieved from experimental data, for this particular section the information was extracted from Gerz et al. [17].

**Table 4.** Types of airplanes according to ICAO categorization. MTOW: Maximum Take-Off Weight.

| | |
|---|---|
| **L (Light)** | MTOW of 7000 kg or less |
| **M (Medium)** | Less than 136,000 kg and more than 7000 kg |
| **H (Heavy)** | 136,000 kg or more |
| **Super Heavy** | Airbus A380-800, with a MTOW of 560,000 kg |

The dimensions and physical values of four airplane models are stated in Table 5; initially no crosswind will be considered. For all four cases, the temporal variation of $\Gamma^*$ will be obtained from experimental data. Thus, from the mentioned references in Table 5, a 6th grade polynomial regression was performed, in order to introduce the temporal variation of the circulation into the in-house program. In all cases, the measurements were made with a LIDAR sensor during the landing procedure.

**Table 5.** Values with which the study was carried out. The density is supposed as equal to $\rho = 1.17 \, \text{kg/m}^3$ and the ambient crosswind as equal to 0 m/s. The initial values needed by the in-house program are the initial generation height, the initial circulation, the wingspan, and the mathematical temporal variation of the circulation. Additionally, the nondimensional initial time is included, since it is necessary to understand the different temporal variation figures of the variables that are presented in the next subsection.

| | Medium Airplane | Heavy Airplane | Heavy Airplane | Heavy Airplane |
|---|---|---|---|---|
| Initial height (m) | 45 | 45 | 47.35 | 94.7 |
| $\Gamma_0$ (m$^2$/s) | 250 | 417 | 458 | 458 |
| $B$ (m) | 34.1 | 57.3 | 60.3 | 60.3 |
| $t_0$ (s) | 18.0 | 30.5 | 30.8 | 30.8 |
| Equation | (36) | (35) | (33) | (34) |
| Used reference | [7] | [12] | [11] | [11] |

The values of the medium airplane correspond to an A320, which receives this categorization according to reference [7]. On the other hand, for the heavy airplanes, the second column has similar values to a Boeing B787-8 and, finally, the third and fourth columns correspond to an A340-300.

The equation of the heavy airplane's circulation of the second column was extracted from LIDAR measurements that were retrieved from Körner et al. [12], whose representation fits Equation (35). On the other hand, the circulation's temporal variation corresponding to the medium airplane was retrieved from reference [7], giving equation number (36).

$$\Gamma^*_{BMA} = 4.87 \cdot 10^{-5} t^{*6} - 9.17 \cdot 10^{-4} t^{*5} + 5.33 \cdot 10^{-3} t^{*4} - 2.77 \cdot 10^{-3} t^{*3} - 0.0592 t^{*2} - 0.0264 t^* + 1 \tag{35}$$

$$\Gamma^*_{A320} = 6.66 \cdot 10^{-5} t^{*6} - 1.59 \cdot 10^{-3} t^{*5} + 1.31 \cdot 10^{-2} t^{*4} - 3.79 \cdot 10^{-2} t^{*3} - 1.74 \cdot 10^{-3} t^{*2} + 1.06 \cdot 10^{-2} t^* + 1 \tag{36}$$

The first results that are presented are the velocity and the pressure fields. A drawback associated with these results, (which were generated for the initial time step), is that the fluid viscosity has no influence on them. However, as already mentioned, we tried to simulate the viscosity effect with the temporal variation of the vortices' circulation, which is specific to each case, see Table 5.

*4.1. Velocity Distribution*

The potential flow equations presented in Section 2 give the value of the physical variables at every point; thus the flow behavior over the entire domain due to the vortices' presence can be obtained.

In order to see a detailed representation of the flow parameters, the figures only picture the results of the right-hand side of the domain, and then the flow values at the left-hand side are symmetric.

Figure 5, which was generated from Equations (14) and (15), represents the velocity field over the selected domain. Due to the circulation sign (positive in this case because the airplane's wing generates lift), the velocity vectors point downwards near the vertical axis, where the distance between the two real vortices, the one shown in Figure 5 and its symmetric, is at a minimum. At the outer part of the vortical structure, the velocity vector of the right-hand side vortex points upwards, but as the velocity vector induced from the symmetric vortex points downwards, the absolute value of the velocity magnitude is lower than at a location near the vertical coordinates' axis.

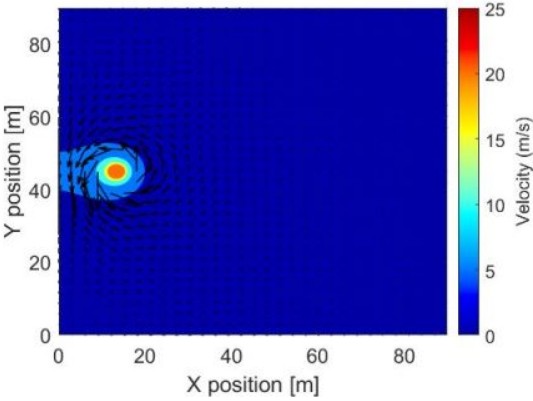

(**a**) Medium airplane; $\Gamma_0 = 250$ m$^2$/s; $U_\infty = 0$ m/s; $h_0 = 45$ m; $B = 34.1$ m.

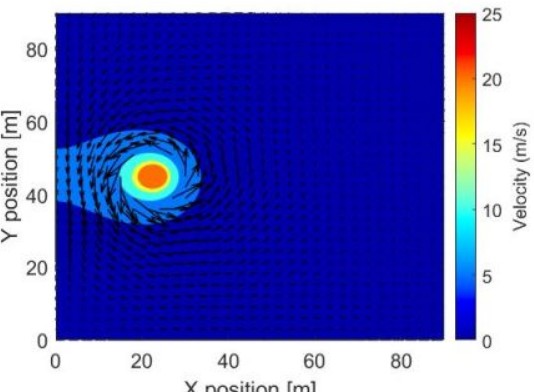

(**b**) Heavy airplane; $\Gamma_0 = 417$ m$^2$/s; $U_\infty = 0$ m/s; $h_0 = 45$ m; $B = 57.3$ m.

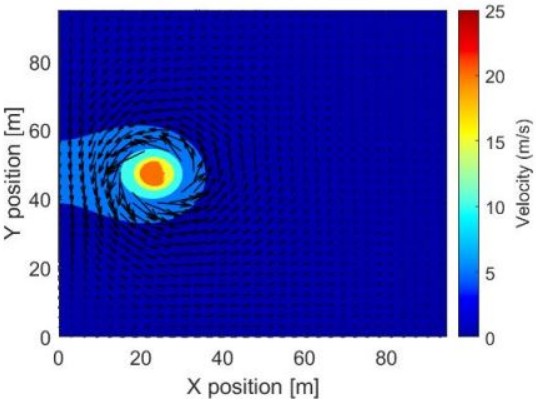

(**c**) Heavy airplane; $\Gamma_0 = 458$ m$^2$/s; $U_\infty = 0$ m/s; $h_0 = 47.35$ m; $B = 60.3$ m.

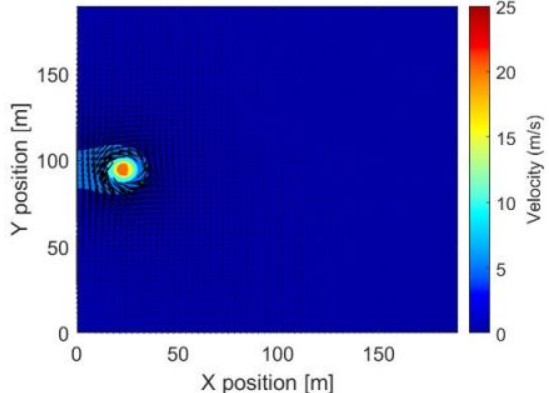

(**d**) Heavy airplane; $\Gamma_0 = 458$ m$^2$/s; $U_\infty = 0$ m/s; $h_0 = 94.7$ m; $B = 60.3$ m.

**Figure 5.** Velocity distribution over the domain for the four cases defined in Table 5 and when ambient crosswind is null. The velocity at every point of the domain has been calculated to later create the vector field.

As expected, and according to the definition of a free vortex, the fluid velocity magnitude increases near the center. Nevertheless, the core of the vortex is represented with constant velocity in Figures 5 and 6. This has been done to better visualize the variables in the graphs.

The dimension of the vortices' central core was retrieved from the definition introduced in Equation (37), which is accepted as a general norm to avoid the asymptote that appears at the vortex center; this equation was used to obtain a better visualization of the variables' change in Figures 5 and 6. To calculate the central core dimension, one needs to know the initial value of the vortices' position in the wingspan, which is presented by $B_A$; such a position is usually given as $B_A = \frac{\pi}{4}B$. The parameter B stands for the wingspan and was initially defined in Tables 1 and 3.

$$r_c = 0.052 B_A = 0.052 \cdot \frac{\pi}{4} B \qquad (37)$$

Figure 5 has been obtained by performing the domain's discretization (where each cell has a size of $\Delta x = \Delta y = 0.1$ m), and calculating the velocity and the pressure at each cell. Then, specifically for Figure 5, the streamline field was obtained with MATLAB's specific function.

For the case of the medium airplane (Figure 5a), the extent of the perturbation generated by the vortex is smaller when compared to other cases. This is due to its lower initial circulation.

On the other hand, in Figure 5b,c, there are inappreciable changes, because both circulation and wingspan are very similar. However for Figure 5d, although it has both initial conditions similar to the previous two figures, the initial height is double, and so its perturbation barely reaches the ground.

Figure 5 also shows the velocity field colored with the same scale for all circulations presented. It can be observed how the zone perturbed by the vortex increases as the circulation rises; notice as well that the velocity at the vortex's central core is maximum. A particularity of the selected discretization of the domain can be clearly seen in Figure 5a,b, where despite the circulation being higher in the case of Figure 5b, it seems that the velocity at the central core of the medium airplane, shownFigure 5a, is higher. This is because the central core radius in the second case is larger due to the higher dimensions of the airplane, as stated in Equation (37). Almost identical phenomena can be seen in Figure 5c.

In order to visualize the dimensions associated with the inner and outer zones and how fast the fluid velocity varies as it gets close to the vortex's central core, the tangential velocity induced by each vortex along a horizontal axis passing through the vortex center is presented in Figure 6, whose data were also retrieved from Equations (14) and (15). As the circulation increases, the maximum tangential velocity keeps rising. The vortex location along the abscissa axis is defined by the airplane dimensions: bigger airplanes have associated larger distances between vortices. (Note that the initial separation $b_0$ is given by $B\pi/4$, where B is the airplane's wingspan).

Regardless of the circulation or the airplane dimensions, the zones where the velocity field suffers major variations are the left and the middle central core, as it is the location where the circulation has a bigger impact. This is because in the zone between the two vortices, the influence of the circulation sums up, and then the induced velocity generated by the two vortices has the same direction. As the circulation increases, the velocity in these zones rises, therefore generating the abovementioned differences.

In Figure 7, we represent the vertical velocity along a horizontal axis situated at the generation height of the vortex. The curves represented in this figure present a similar behavior to the one explained in Figure 6, but now the sense of the velocity vector can be clearly seen. The positive and negative peaks can be understood when considering the counter-clockwise rotation of the starboard vortex.

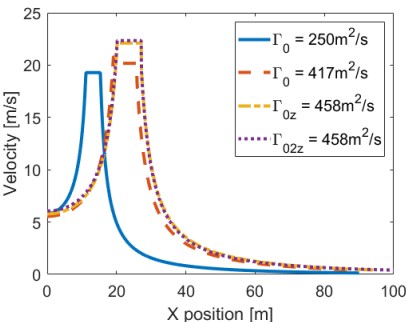

**Figure 6.** Tangential velocity induced by the vortex when no crosswind is considered. Velocity is calculated at the vortices' generation height. The circulation values are taken from Table 5.

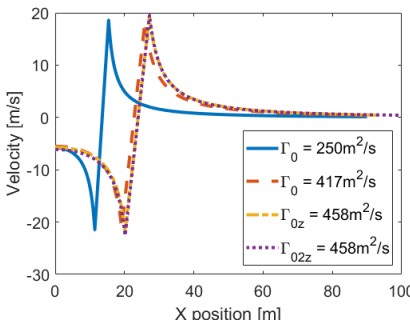

**Figure 7.** Vertical velocity induced by the vortex when no crosswind is considered. Velocity is calculated at the vortex's generation height. The circulation values are taken from Table 5.

Based on Equations (33)–(36), Figure 8 shows the temporal variation of the circulation. These temporal variations, as has already been explained, were retrieved from experimental measurements performed through LIDAR sensors, whose results are stated in the corresponding references established in Table 5. This figure is presented to have an idea of the circulation evolution. In all cases, the temporal decay is minor at early times, but then a rapid phase decay occurs, which is usually given, at non-dimensional time $t^* = 2$, where the circulation falls down rapidly. Finally, the beginning of the second phase normally coincides with the apparition of the vortex rebound on the vertical axis.

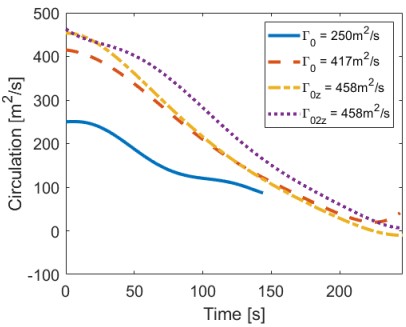

**Figure 8.** Temporal variation of the vortex circulation when no crosswind is considered. These curves were obtained from the references presented in Table 5.

### 4.2. Pressure Distribution

The main characteristics of the pressure distribution over the domain and for the different circulations studied were obtained by means of Equation (1) and are presented in Figure 9. The pressure is minimum at the vortices' center due to the high turning speed associated with the flow. The pressure variation near the vortices' center is particularly drastic. This big adverse pressure gradient in reality causes a boundary layer detachment,

and according to Asselin et al. [2] a secondary vortex is generated, which causes the first one to slow down and to rebound once it gets near the ground.

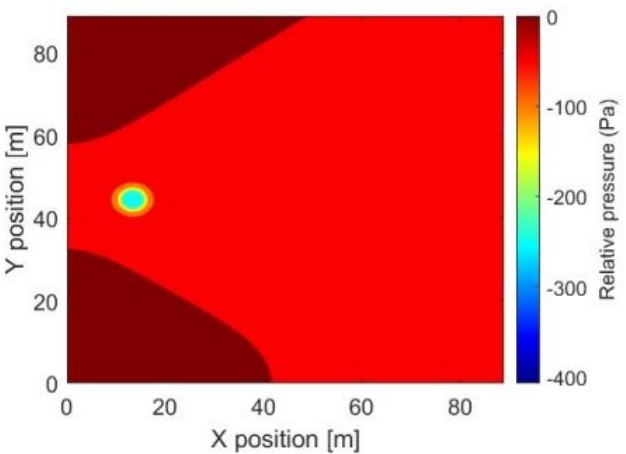

(**a**) Medium airplane; $\Gamma_0 = 250 \text{ m}^2/\text{s}$; $U_\infty = 0 \text{ m/s}$; $h_0 = 45 \text{ m}$; $B = 34.1 \text{ m}$.

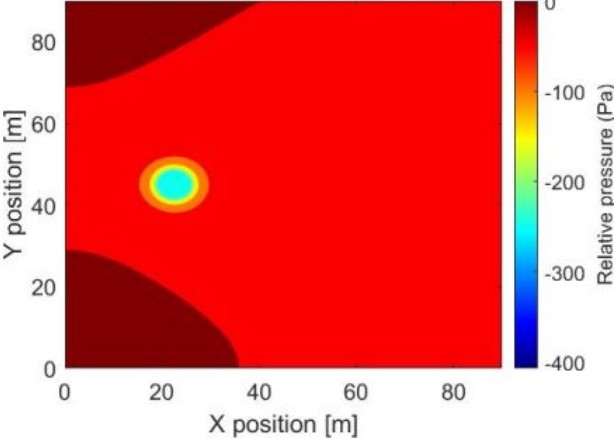

(**b**) Heavy airplane; $\Gamma_0 = 417 \text{ m}^2/\text{s}$; $U_\infty = 0 \text{ m/s}$; $h_0 = 45 \text{ m}$; $B = 57.3 \text{ m}$.

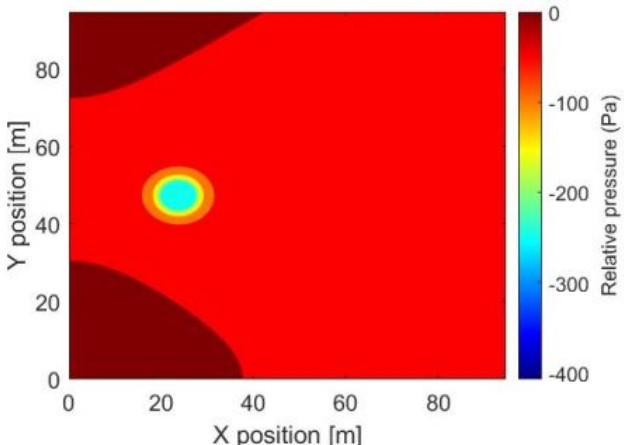

(**c**) Heavy airplane; $\Gamma_0 = 458 \text{ m}^2/\text{s}$; $U_\infty = 0 \text{ m/s}$; $h_0 = 47.35 \text{ m}$; $B = 60.3 \text{ m}$.

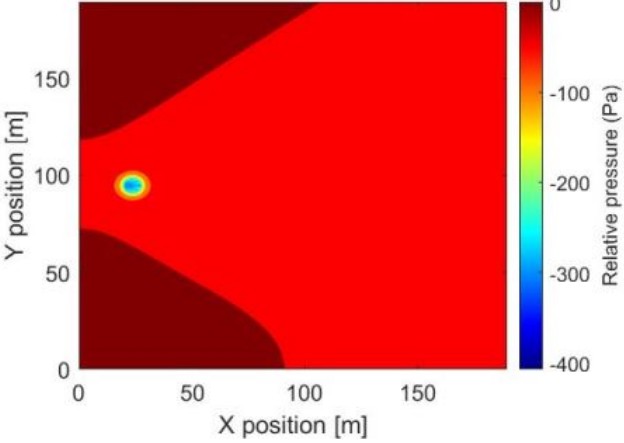

(**d**) Heavy airplane; $\Gamma_0 = 458 \text{ m}^2/\text{s}$; $U_\infty = 0 \text{ m/s}$; $h_0 = 94.7 \text{ m}$; $B = 60.3 \text{ m}$.

**Figure 9.** Pressure distribution over the domain when there is no crosswind. Equation (37) is used to obtain the dimension of the vortex central core. The domain discretization employed is identical to the one used in Figure 5.

This effect is far from the ideal one, where the viscosity is supposed to be null and in which the vortex follows a hyperbolic trajectory. In order to obtain the vortices' evolution as near as possible to the inviscid case and as previously mentioned in the Introduction, Wakim et al. [3] propose a method consisting of sucking the boundary layer formed near the ground. If this method could be physically implemented in the future, the vortices would follow an ideal hyperbolic trajectory, like the ones presented in this paper.

One aspect that weighs down the pressure graphic is that since the pressure variations near the vortices' central core are large, the details over the domain cannot be clearly seen. However, as it happens with the velocity field, the area affected by the pressure field (see Figure 9) increases with the circulation increase.

*4.3. Vortex Displacement*

Another feature that complements the created in-house program is the fact that the ideal trajectory of the vortex can be obtained based on Equations (24) and (25) combined with the time elapsed. In Figure 10 it is seen that when the vortices are close to each other,

the main displacement happens versus the Y-axis; this is because the vertical downwards velocity of its center is reasonably high, but after some time the only remaining velocity is the horizontal one, and then the vortex travels parallel to the ground.

The passage from a vertical movement to a horizontal one is due to the presence of the ground. In Figure 10 one can see how the vortex with a higher initial height starts moving in the vertical coordinate and only changes its motion to the horizontal axis at the same point of the vortices whose initial circulation is the same, although their initial height is lower. It can therefore be concluded that the horizontal velocity is directly affected by the presence of the ground.

Finally, looking at the medium aircraft's vortex, its motion along the Y-axis is bigger than for the other cases, because the two vortices generated by the airplane are closer together, meaning that its influence is higher, so the induced vertical velocity from one to the other will also be higher. This effect pushes the vortices further down and the horizontal movement starts at a smaller distance to the ground.

In Figure 11 based on Equation (25), although the initial height of the vortices of the different airplanes is nearly the same, it is observed how at the early stages of the study, the vortices' vertical velocity displacement of the medium airplane is bigger (in absolute value) than the one of the larger airplanes. The acceleration is larger as well in the case of the medium airplane; notice that the slope of the velocity curve is steeper. This higher acceleration is because for the medium airplane the two vortices are closer than for larger ones. After about 60 s the vertical velocity of the vortices tends asymptotically to zero.

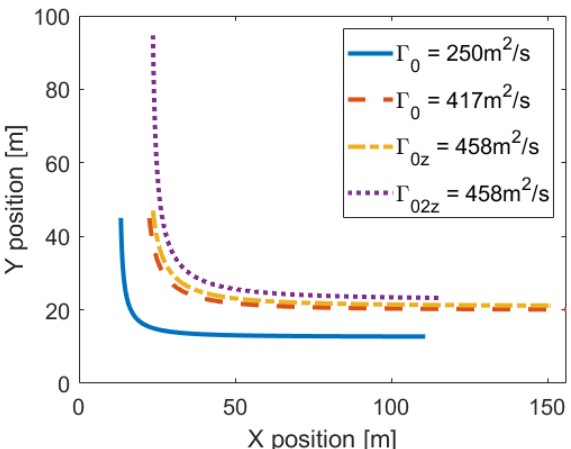

**Figure 10.** Variation of the vortex center's position throughout time, when no crosswind is considered.

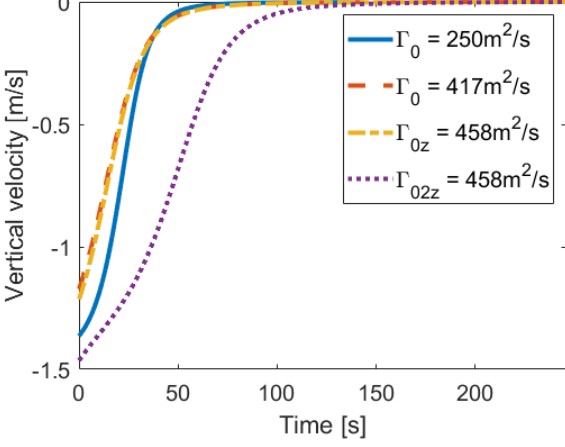

**Figure 11.** Vertical velocity of the vortex's center, when no crosswind is considered.

On the other hand, for the vortices generated at a higher altitude, the initial velocity is the highest among all (in absolute value). Additionally, as the time increases, its pendent becomes steeper and the vertical acceleration increases. Finally, as has been already shown in Figure 10, the vertical velocity takes longer to become null in this case. Once the separation is high enough, the vertical velocity becomes 0, because the influence of the vortices between them becomes null.

When considering the temporal evolution of the horizontal velocity Equation (24), see Figure 12, it increases with time until reaching a maximum value and then steadily decreases tending to zero, providing the computational time is long enough. For the case of the medium airplane, the horizontal velocity does not decrease uniformly, this may be due to oscillations associated with the circulation, and then in Figure 8 the circulation of this particular airplane is the one that decreases less steadily. Additionally, the peak of the airplane with circulation $\Gamma_{02z}$ is lower than the other ones; this is because, when this vortex arrives close to the ground, its circulation has hugely decreased, and thus the ground influence is lower and as a result the horizontal velocity will not be as high as the one associated with the other airplanes.

Such a velocity decrease appears reasonable; thus as the vortices separate, their mutual influence reduces. From Figure 12, it is also observed that the horizontal velocity is directly affected by the presence of the ground. Because even though the vortex of the medium airplane has a lower circulation than the heavier airplane's vortices, and at the initial time the circulation value is smaller, due to its faster approach to the ground, the horizontal acceleration during the initial time steps is higher in this case. In addition, this causes its peak to become like the other ones, despite the previously mentioned characteristics of the initial conditions.

Finally, from Figure 12 it is also seen that the maximum value of the horizontal velocity keeps increasing with the circulation increase, providing that the initial height is the same. The peak of the horizontal velocity coincides with the time when the vertical velocity starts to be close to 0. Taking into account that the horizontal velocity is directly affected by the ground effect, this time characterizes the moment the vortex is closest to the ground, but also the two real vortices are still close enough to maintain an intense influence on each other. Notice that this time reduces as the circulation increases, due to the bigger mutual influence (in this case it can be hardly seen because both heavy airplanes with the same initial height have similar initial circulations). Notice that the medium airplane's horizontal velocity arrives at its peak earlier than any other due to the closer position of the vortex to the ground.

In order to evaluate how long it takes until the vortices move away from the runway, Figure 13 is used. As a consequence of the velocity values presented in (Figures 11 and 12), it is observed that the rate of separation is smaller at the initial time because the vertical velocity has a bigger influence on the vortices' displacement than the horizontal one.

The differences in the separation curves between the two heavy airplanes with the same initial height observed in Figure 13 are almost negligible, and in fact they are due to the temporal variation of the circulation. A crossover point at nearly 180 s is observed and is due to the faster separation of the vortices with $\Gamma_0 = 417 \text{ m}^2/\text{s}$ than the ones of the other airplane. The separation evolution curve for the medium airplane seems to be parallel to the previous two in the early stages (due to its horizontal velocity and acceleration at early times), although after some time the curve tends to separate at a lower rate. The evolution for the final time steps seems to be more linear that the previous two. Finally, the vortices whose rate of separation is the slowest are the ones generated by the heavy airplane at a higher altitude, due to the previously commented upon characteristics regarding its horizontal velocity. The vortices do not separate until about 50 s and when they come close to the ground they start to move horizontally. As soon as the horizontal velocity reaches its maximum and starts to decrease, the rate of separation also decreases tending to an asymptotic value.

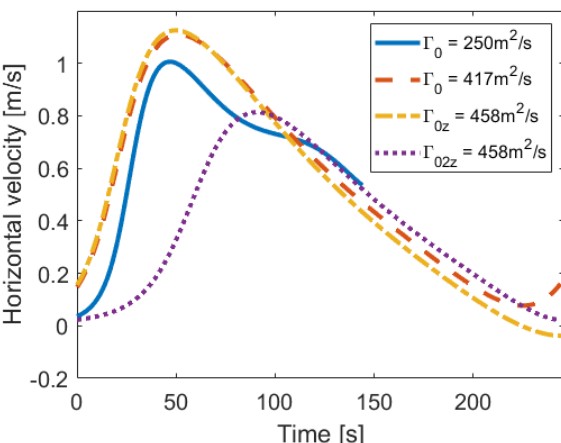

**Figure 12.** Horizontal velocity of the vortex center when ambient crosswind is considered null.

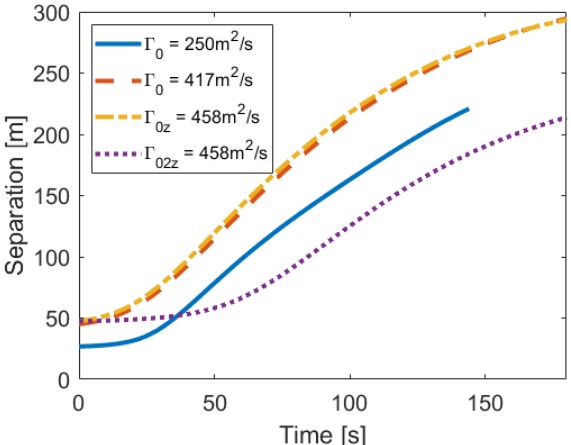

**Figure 13.** Separation between the vortex centers when crosswind is considered equal to 0 m/s.

Taking into account that the widest runway measures 60 m, the vortices introduced in Figure 13 should be out of the runway in less than 40 s (in the case of the medium airplane and the heavy airplanes with an initial height of 45 m), and in less than 70 s for the case of the vortices generated by a heavy airplane at an initial height of 97 m. Nevertheless, the time the following plane should wait until facing the runway should be longer than the one just described, as even though the centers of the vortices are expected to be outside the runway, their influence is expected to take longer to disappear; this point is discussed in Section 4.7.

### 4.4. Vortex Movement and Circulation under Crosswind Conditions

Having analyzed the case without crosswind, the next step is to evaluate the effect of the crosswind on the vortical structures.

In order to validate the in-house program when crosswind is included, a comparison between the results obtained with the present mathematical model and the ones obtained by means of LIDAR measurements, taken from Holzäpfel [11], is presented. The procedure to obtain the temporal variation of $\Gamma$ is the same as the one explained in Section 3. The equations employed to characterize the circulation decay are Equations (33) and (34).

As was previously explained, to generate Figure 14, the figures belonging to Holzäpfel [11] were extracted from the article and then modified to overlay the circulation given by the polynomial regression done in the in-house program.

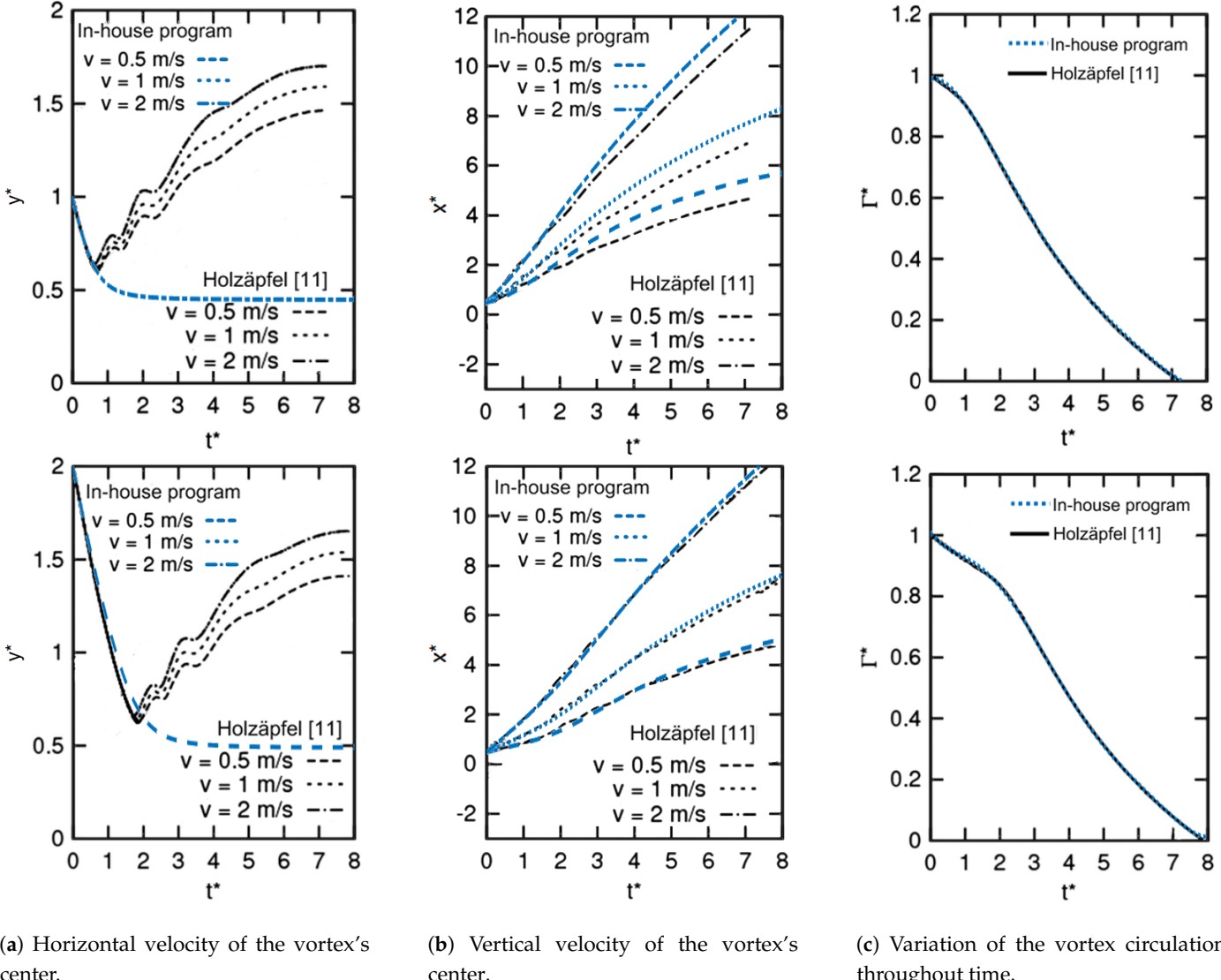

(**a**) Horizontal velocity of the vortex's center.

(**b**) Vertical velocity of the vortex's center.

(**c**) Variation of the vortex circulation throughout time.

**Figure 14.** Comparison between the present study's results and the experimental results obtained from reference [11] (under different crosswind conditions, at 0.5, 1 and 2 m/s). The different figures belonging to the mentioned reference were edited to later overlay the prediction obtained with the in-house program. In all cases, the experimental results from Holzäpfel [11] are plotted in black color. The three upper figures characterize the results at $z = b_0$ (whose results were obtained with the circulation temporal variation shown in Equation (33), and the ones at the bottom represent the case at $z = 2b_0$ (following Equation (34) for the temporal variation of circulation).

The behavior of the vortices is almost the same, as seen in Section 3. In the case of the horizontal evolution $x^*$, the values for $y_0 = 2b_0$ are very accurate. For the lower initial height $y_0 = b_0$, the accuracy of the results is acceptable, although the position of the vortex center along the y* axis is slightly overestimated. This could be due to the absence of the boundary layer in the potential flow theory, meaning that the vortex motion is not affected by it. Finally, it seems that the error is lower for higher crosswind velocities, $U_\infty$.

On the other hand, when considering the vertical position evolution $y^*$, the values are more accurate for the case of $y_0 = 2b_0$. But in both cases, the prediction becomes loose when the vortex rebound phenomena occur. One characteristic that must be noted is that the vertical position is independent of time in the in-house program—a fact that could not be taken as valid, because as $U_\infty$ increases, the vortex rebound is bigger.

In order to have yet another comparison, the data from article [12] were used. There, the results of the in-house program, using a two-phase wake vortex decay circulation, are compared to a Bayesian Model Averaging (BMA) method. This method provides a slightly different circulation of decay throughout time (Figure 15c), and thus it will be useful to compare the potential flow model to another decay method. The $\Gamma$ evolution throughout time, in this case, fits the previously stated Equation (35).

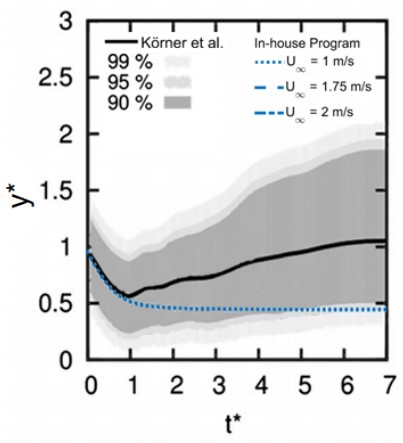

(**a**) Horizontal velocity of the vortex's center.

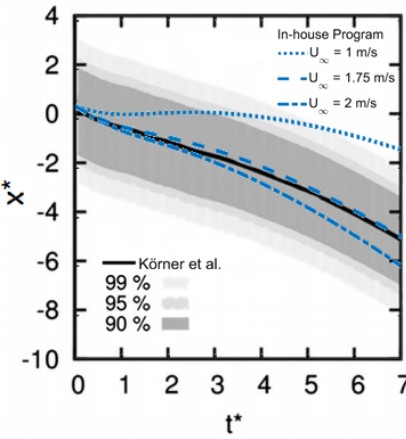

(**b**) Vertical velocity of the vortex's center.

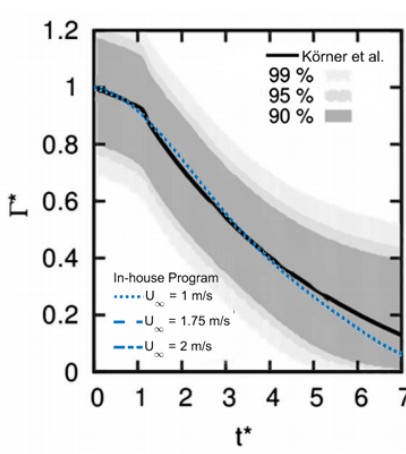

(**c**) Variation of the vortex circulation throughout time.

**Figure 15.** Comparison between the present study's results and the experimental results obtained from Körner et al. [12] (under crosswind conditions, at an unspecified velocity). To perform the comparison, figures belonging to the mentioned reference were extracted from the article, and then modified to overlay the data given by the in-house program. Additionally, the probabilistic envelopes have been included to see the precision of the in-house program.

In Figure 15a, the in-house program correctly predicts the behavior at early times, but after $t^* = 1$ it presents a deviation. Additionally, at approximately $t^* = 5$ it abandons the 90% envelope. To be noted is that for the potential flow model the vertical position of the vortex center is not affected.

In Figure 15b, a comparison between the model presented by Körner et al. [12] and the in-house program for different horizontal velocities is performed. In the figure it is observed that the experimental study has a perfect agreement with the mathematical model when considering a crosswind speed of 1.75 m/s. However, the agreement is also very good when the crosswind velocity is of $U_\infty = 2$ m/s. Both crosswind velocities are inside the 90% probability envelope, from which it can be concluded that the mathematical model presented is very accurate.

Before presenting the rest of the graphics that complete the analysis under crosswind conditions, Table 6 must be introduced in order to understand the legend of the following figures.

**Table 6.** Abbreviations used in the legend of following figures.

| Kind of Airplane | $\Gamma_0$ (m²/s) | $U_\infty$ (m/s) | Initial Height (m) | Abbreviation |
|---|---|---|---|---|
| Medium airplane | 250 | 0 | 45 | 250-0 |
| Medium airplane | 250 | 1 | 45 | 250-1 |
| Medium airplane | 250 | 2 | 45 | 250-2 |
| Heavy airplane | 458 | 0 | 47.35 | 458-0 |
| Heavy airplane | 458 | 1 | 47.35 | 458-1 |
| Heavy airplane | 458 | 2 | 47.35 | 458-2 |
| Heavy airplane | 417 | 2 | 45 | 417-2 |
| Heavy airplane | 458 | 2 | 94.7 | 458-2-h2 |

Now, a significant characteristic to be kept in mind is the fact that regardless of the crosswind velocity, the separation between vortices remains the same. This is because potential flow is employed in both cases, with and without crosswind. Therefore, the crosswind will move the two vortices altogether but will not increase the separation between them. Nevertheless, this behavior is very similar to the real case, as is stated in reference [13]. Then, the vortices will have the same separation as the one shown in Figure 13, no matter which crosswind is applied. This is because as can be extracted from Equation (14), once the subtraction between the velocity of the left and right vortices is done, the $U_\infty$ value will disappear, and the rate of separation will be only a function of the circulation and the position of the vortices. The difference when adding up a $U_\infty \neq 0$ with respect to the unperturbed case can be seen when comparing Figures 16 and 17. Both were obtained by combining Equations (24), (25) and (34) with the elapsed time. In these two figures, the trajectory of the vortex pair can be observed, as well as how it is changed when crosswind appears, moving them further sideways with the wind's direction. This movement is explained thanks to Equation (24), as the only change due to the appearance of the crosswind is that the value of $U_\infty$ is not 0 for any of the two vortices. Although in Figure 17 it seems that the separation between vortices is lower than the one in Figure 16, in reality, it is an effect produced by the horizontal scale of Figure 17; then, when observing the position between the starboard (right side of the airplane) and the port (left side of the airplane) vortices in the last time step, the separation in both cases is the same (from, approximately, $-115$ to $115$ m in Figure 16, and from, approximately, $375$ to $605$ m in Figure 17).

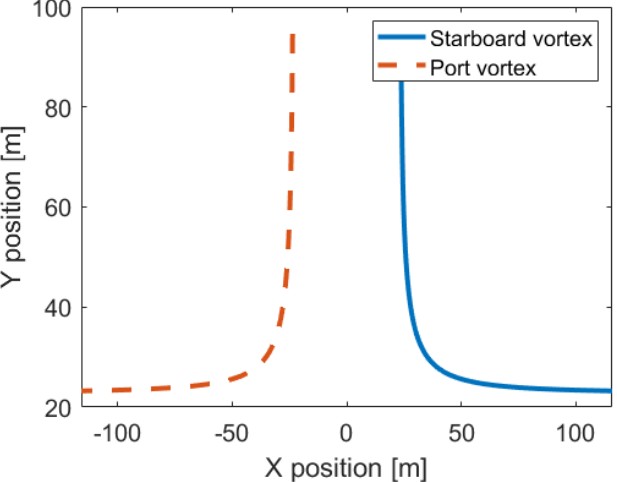

**Figure 16.** Vortices' centers' positions throughout time with $U_\infty = 0$ m/s.

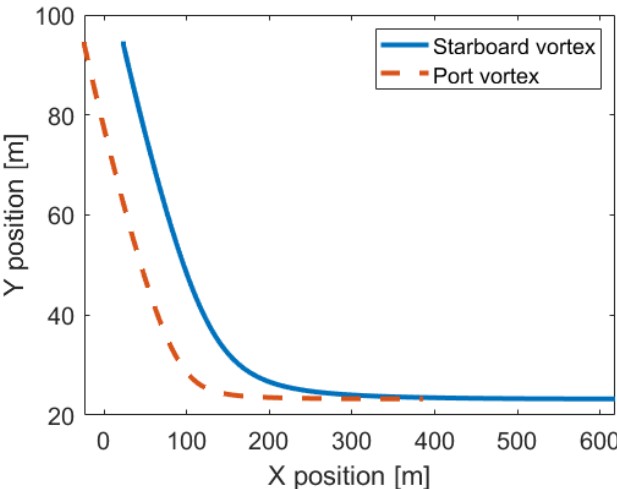

**Figure 17.** Vortices' centers' positions throughout time with $U_\infty = 2$ m/s.

In addition, due to the crosswind effect, the velocity field suffers a modification, therefore affecting the pressure field, and then both fields have a direct relation as stated in Equations (1) and (28). The main characteristic in the relative pressure field over the ground (which is obtained by fixing the Y-axis position to 0 in Equation (1) is that as the crosswind speed increases and the minimum pressure peak moves towards the left, see Figure 18. It can also be seen that as $U_\infty$ increases, the initial pressure drop keeps growing and the pressure recovery along the abscissa axis becomes smaller. In other words, the typical pressure recovery observed when there is no crosswind keeps disappearing as the crosswind effect is considered. In fact, the relative pressure in the right part of the domain remains negative when the crosswind effects are considered. This information is directly extracted from Equation (1). It is also to be noticed that at a certain distance from the vortex's central core, the pressure and flow fields are no longer influenced by the vortices' effect; the only perturbation is given by the $U_\infty$, and pressure and flow fields will just depend on the crosswind.

Finally, looking at the different cases presented in Figure 18, the case whose pressure drop is bigger is the one of the heavy airplane with a higher initial position; additionally, this peak is the one that has moved further to the right. Among the other airplanes, the biggest pressure drop is given by the medium airplane, because it gets closer to the ground and the pressure and velocity fields are particularly influenced by it.

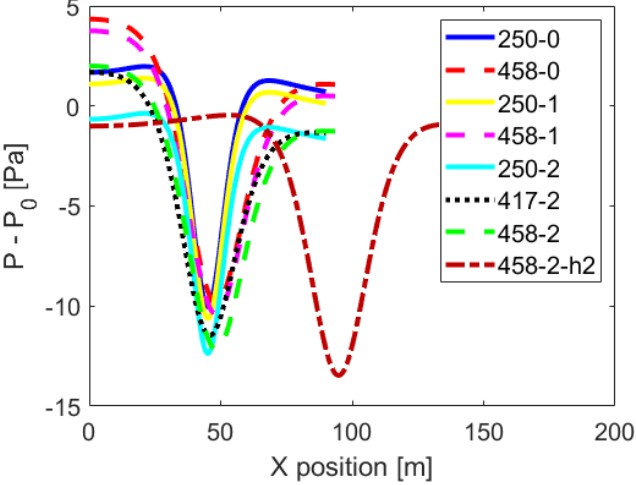

**Figure 18.** Comparison of the relative pressure variation over the ground when different crosswinds (From $U_\infty = 0$ m/s to $U_\infty = 2$ m/s) are applied to different airplane models (Table 6).

Another parameter that is not affected by the crosswind is the temporal variation of the circulation; as it be seen from Equations (33)–(36), its value is independent from the crosswind velocity. This means that all of the cases with the same initial $\Gamma_0$ will also have the same temporal variation. This is why Figure 8 will stay true for all crosswind cases.

The temporal variation of the vortex position when the crosswind velocity effect is considered is shown in Figure 19. It can be observed that as the crosswind velocity increases the vortex is further displaced sideways, but the final vertical position remains unchanged. This is a direct effect of the vertical and horizontal velocity, because even though the vertical velocity is not dependent on the crosswind, the horizontal is. This means that as the crosswind speed increases, the eccentricity of the hyperbola will be higher. From this figure it can also be extracted that the movement along the Y-axis is a function of the airplane's dimension rather than of the circulation. As is seen in Figure 11, the vertical velocity for the two different circulations of the heavy airplane (with the same initial height) is almost the same, and therefore the final Y-axis position in these cases will be also nearly the same. Yet, due to the temporal variation of $\Gamma$, the vortices with higher circulation will end their path in a slightly higher position in comparison with the ones with a lower circulation. This small change can be produced because the initial separation between vortices is lower for the case of lower circulation, which can generate a bigger interaction between them. Additionally, in the case that the initial height of the vortex is higher, the final position will also be slightly higher, compared to those vortices with the same initial circulation but initialized at a lower height. A lower circulation means that the vertical velocity induced by the vortices themselves is smaller, and thus they will move slower and stop the movement earlier. However, this last statement only applies to an airplane with the same dimensions, because as seen in Figure 19, even though the circulation of the medium airplane is smaller than the one of the heavy airplane, the final position of its vortex is shorter, approximately 10 m less when compared with the position of the vortex generated by a heavier airplane.

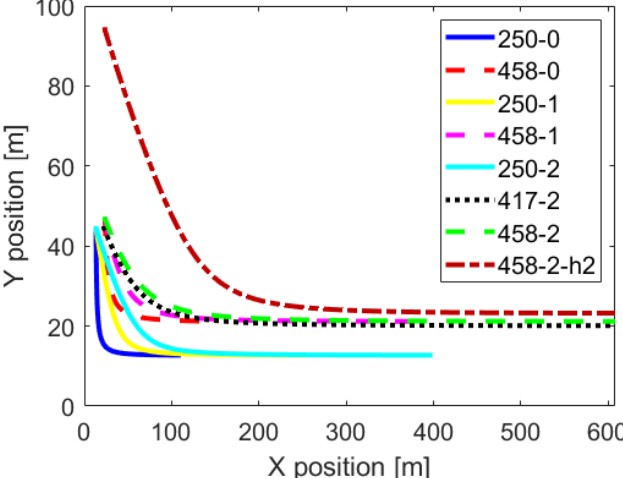

**Figure 19.** Comparison of the starboard vortex center's positions throughout time, when different crosswinds (from $U_\infty = 0$ m/s to $U_\infty = 2$ m/s) are applied to different airplane models (Table 6).

Once the motion of the vortices is introduced, it is interesting to see how their velocity varies throughout time. First, as previously explained, the vertical velocity is independent of the crosswind, as can be seen from Equation (25), and therefore the temporal variation will be the same as in the case without crosswind (Figure 11). This is since the separation between vortices remains the same regardless of the crosswind velocity.

In the case of the horizontal velocity, the only change due to the appearance of the crosswind is that this value will be added up directly to the value already calculated supposing no crosswind, as can be observed from Equation (14). It can be seen in Figure 20

that as the circulation increases, the peak on the horizontal velocity becomes higher. The acceleration of the velocity on the X-axis is higher in the case of the medium airplane relative to the heavy ones. Even though at the initial time the velocity induced in this direction by the vortices themselves is almost 0, this particularly high acceleration is likely to be produced by the effect of the ground on the vortical structures. The behavior of the vortices generated by a heavy airplane at a higher altitude follows the same tendency as the case without crosswind, meaning that the peak is delayed with respect to the other models. Finally, for the medium airplane, the horizontal velocity becomes higher than the one of the heavy airplanes at approximately 110 s. This may be either due to the ground effect or due to a perturbation in the temporal variation of the circulation, which can be seen in Figure 8.

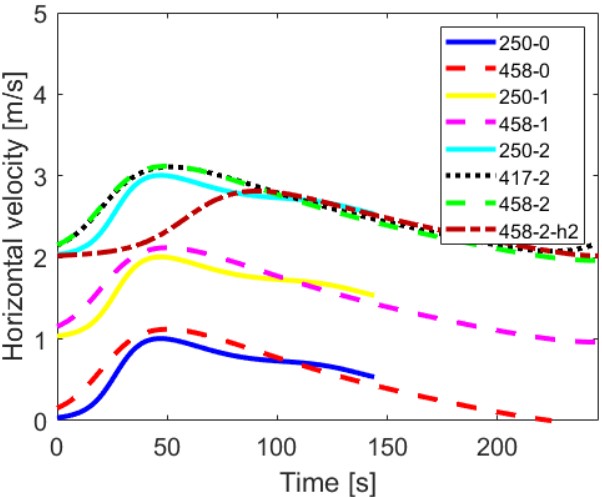

**Figure 20.** Comparison of the horizontal velocity of the vortex center, when different crosswinds (From $U_\infty = 0$ m/s to $U_\infty = 2$ m/s) are applied to different airplane models (Table 6).

Figure 21 represents the velocity field for different circulations and crosswind velocities. In general, when comparing Figure 5 and Figure 21, it is clearly seen how the velocity changes along the domain as the crosswind velocity increases, and the entire flow field is now more intense. Additionally, when crosswind is considered, there are zones where the velocity is null. In fact, for all of the cases studied with crosswind, the velocity is much higher below the vortex than above it; this is because below the vortex the velocity induced by the vortex and the velocity of the crosswind sum up (note the change of the light blue zone), and these two velocities flow in opposite directions above the vortices. Additionally, the area affected by the perturbed flow is wider and more irregular when crosswind is considered. Nevertheless, in both cases most of the domain remains poorly affected and the flow velocity in the nearest zone around the central core remains almost unaffected by the crosswind velocity. This is likely to be due to the large velocity field induced by the vortices, which is much higher in comparison to the induced velocity generated by the crosswind.

The main difference between Figure 5c in respect to Figure 5a,b resides in the intensity of the perturbation to the flow, which is lower in the case of a lower circulation. There is another difference that happens in the zone where the velocity is near 0. This zone moves from the quasi-central upper side to the upper right-hand part of the vortex as the circulation increases. Above this zone, the particles move to the right, as the speed of the vortices in this zone is not strong enough to overcome the crosswind. Within Figure 5a,b just minor differences can be appreciated; nevertheless, the field's intensity in the case of lower circulation seems to be higher, due to the fact that vortices are generated closer to each other, and the influence in the inner part of the vortex is a little bit higher, producing also higher velocities.

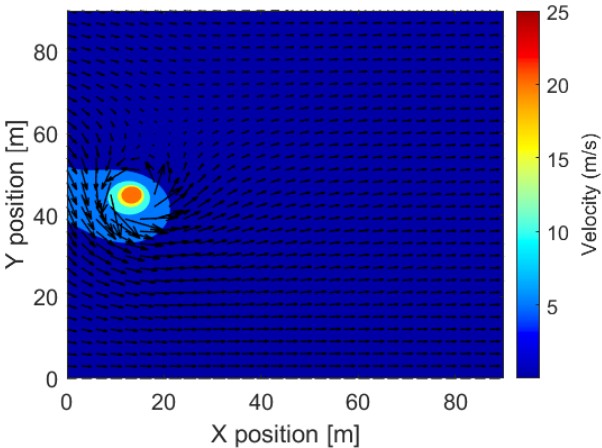

(**a**) Medium airplane; $\Gamma_0 = 250$ m$^2$/s; $U_\infty = 2$ m/s; $h_0 =$ 45 m; $B = 34.1$ m.

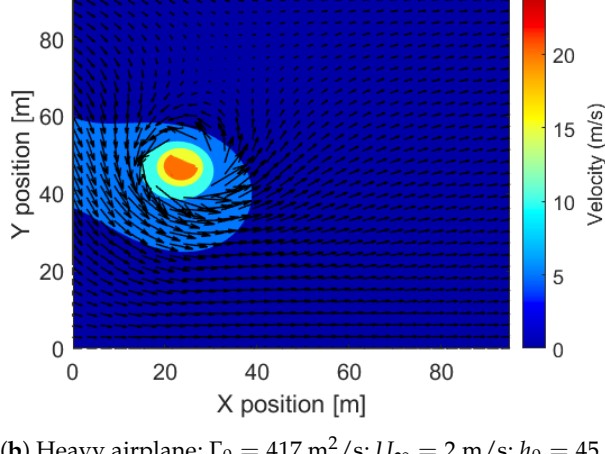

(**b**) Heavy airplane; $\Gamma_0 = 417$ m$^2$/s; $U_\infty = 2$ m/s; $h_0 = 45$ m; $B = 57.3$ m.

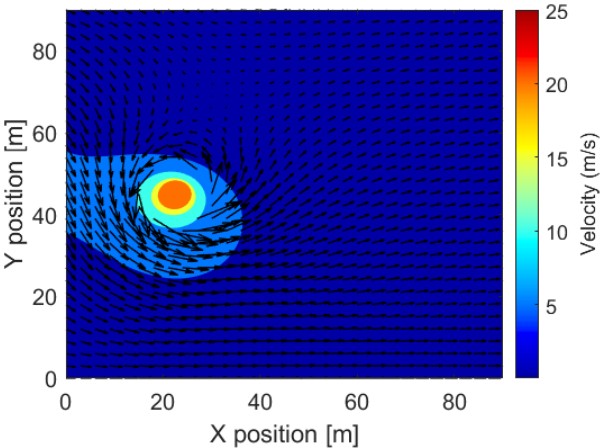

(**c**) Heavy airplane; $\Gamma_0 = 458$ m$^2$/s; $U_\infty = 2$ m/s; $h_0 =$ 47.36 m; $B = 60.3$ m.

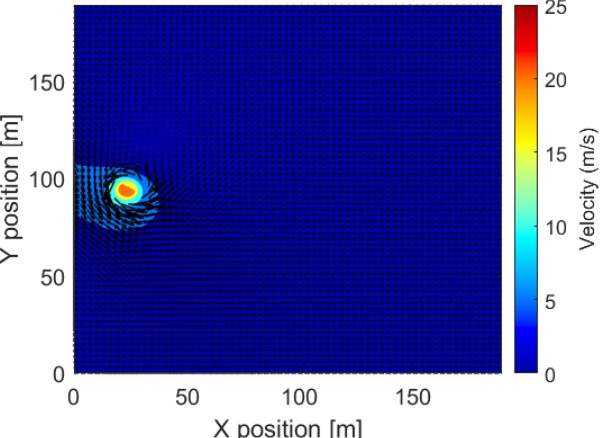

(**d**) Heavy airplane; $\Gamma_0 = 458$ m$^2$/s; $U_\infty = 2$ m/s; $h_0 =$ 94.7 m; $B = 60.3$ m.

**Figure 21.** Velocity distribution over the domain with crosswind. The velocity at every point of the domain was calculated to later create the vector field.

Finally, the differences due to the existence of crosswind are more visual in Figure 21d, especially when compared to Figure 5d. Now the domain is completely affected by the velocity field; in Figure 5d, the influence of the vortex to the flow domain was rather small, due to its initial height. In Figure 21c,d, the velocity field differences in the zone between the vertical axis and the vortex are almost inappreciable, and the ones that appear are likely to be due to the different position to the ground.

When considering the pressure field, see Figure 22, the graphs greatly resemble the ones without crosswind; compare Figures 9 and 22. Whenever the crosswind is considered, the pressure and flow fields lose their symmetry, the pressure at the entire field tends to decrease, and when looking at the central core of the vortices, the symmetry is also lost.

Comparing Figure 22a,b, the main difference is that the zones with a pressure nearer to 0 reduce their area as the crosswind speed increases. Additionally, pressure symmetry around the central core is lost with the crosswind velocity increase, and the pressure below the vortical structure tends to reduce.

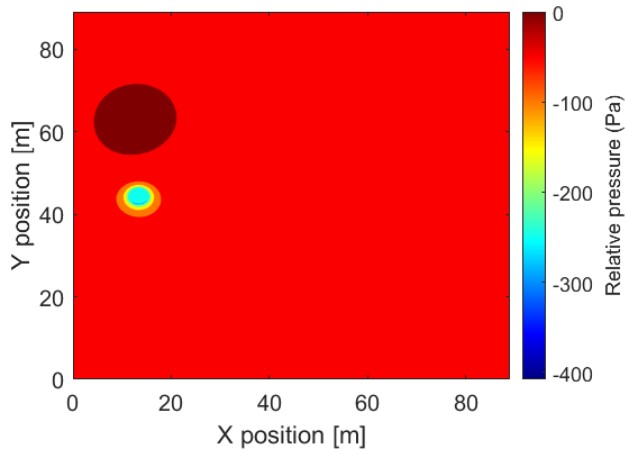

(**a**) Medium airplane; $\Gamma_0 = 250$ m$^2$/s; $U_\infty = 2$ m/s; $h_0 = 45$ m; $B = 34.1$ m.

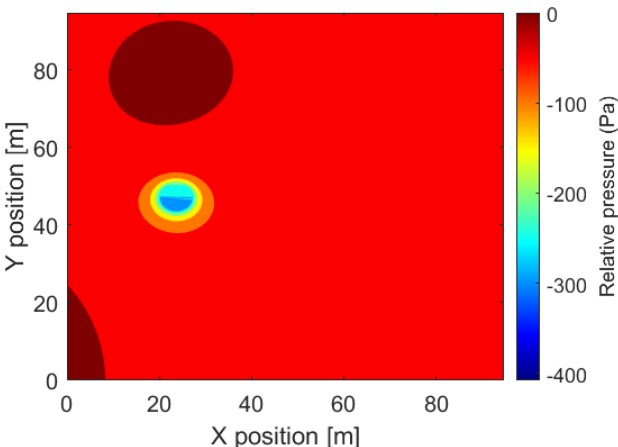

(**b**) Heavy airplane; $\Gamma_0 = 417$ m$^2$/s; $U_\infty = 2$ m/s; $h_0 = 45$ m; $B = 57.3$ m.

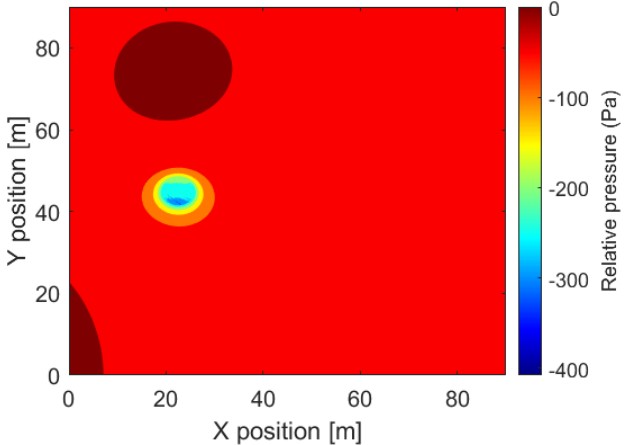

(**c**) Heavy airplane; $\Gamma_0 = 458$ m$^2$/s; $U_\infty = 2$ m/s; $h_0 = 47.36$ m; $B = 60.3$ m.

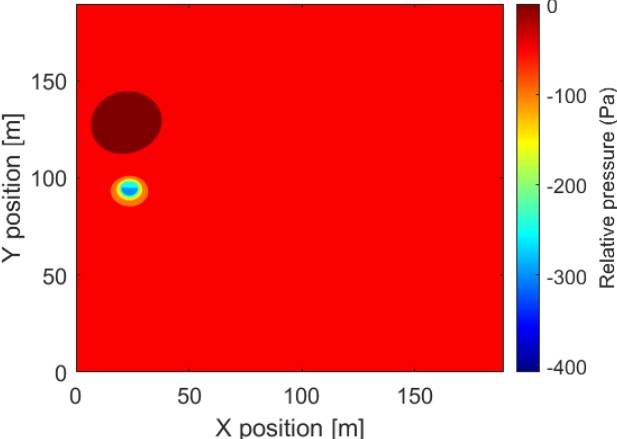

(**d**) Heavy airplane; $\Gamma_0 = 458$ m$^2$/s; $U_\infty = 2$ m/s; $h_0 = 94.7$ m; $B = 60.3$ m.

**Figure 22.** Pressure distribution in the area near the vortex with crosswind.

When comparing the figures with the same crosswind, it is seen that the smaller the initial circulation, the smaller the flow field area perturbed by the vortex. Additionally, the minimum pressure area is smaller when comparing the vortices generated by a medium airplane to the ones obtained from a heavy one (Figure 22a in comparison to Figure 22b,c). Additionally, the bubble over the vortex increases its area and displaces further up as the circulation increases.

Another aspect that can be noted is the fact that the center of the vortex experiences a harder depression as the circulation increases. Nevertheless, between similar circulations, the depression is a function of the distance between the two real vortices at its generation (see Figure 22b,c). Finally, within Figure 22c,d, the depression observed at the center of the vortex is higher for the one generated at a higher altitude.

### 4.5. Flow Field Evolution When No Crosswind Is Considered

After seeing different comparisons between models, it can be concluded that the two-phase wake vortex decay theory, implemented in the potential flow program, provides a good approach towards the evolution of the vortices. In order to visualize the temporal flow behavior around the near vortices domain, Figure 23 was created, which shows the temporal evolution of the particles around the vortex's central core for two different circulations, two different initial heights and when no crosswind is considered. Each one

of the three different cases presented in this figure are characterized by the temporal decay of the vortices' circulation defined by Equations (33), (34) and (36). The analysis of the variation of its near flow field throughout time will be done until a nondimensional time of 2, which is the maximum time at which the in-house program is very accurate.

This figure clearly shows how the central core moves downwards near the ground while increasing the distance between the homologous core located on the left-hand side and not presented in the figure. It is also seen how the central core grows and changes its shape from the initial almost circular one to a nearly oval one after a time step equal to $t^* = 2$. These two aspects are less intense for the case of the vortices generated at a higher altitude (see Figure 23c) due to its greater distance from the ground. The same happens for Figure 23a, but this is due to the fact that its nondimensional time is lower compared to the other ones; see Table 5.

As the circulation increases, the central core grows and the perturbed flow area around the vortex becomes larger. In fact, the vortex evolution shows how the fluid particles tend to rotate together with the vortex. Initially, just the central core near the flow field follows this motion, but as time develops the entire flow domain keeps being affected by the rotation of the vortices' central core. This figure also shows that until a non-dimensional time $t^* = 1$, the vortices simply move downwards, and the central core maintains a nearly rounded shape. At the two final times presented ($t^* = 1.5$ and $t^* = 2$), it is seen how the vortex center is displacing slightly downwards and considerably moving along the abscissa axis, changing the shape to an almost oval one. This deformation is mostly due to the presence of the ground. This situation is less exaggerated in Figure 23c, because at the last time step the vortex is still arriving to the ground, and thus its lateral motion and core deformation are not clearly visible. The behavior of the vortex shown in the third column (Figure 23c) is completely different than in the other two cases, because its generation height is higher. This means that its vertical displacement is longer, as is clearly seen in the corresponding set of figures. The horizontal velocity is lower compared to the other two cases, very likely due to the absence of the ground effect at early stages. Additionally, even though the vortex core increases its dimension, it maintains the round shape for all of the time steps, although at time step $t^* = 2$ its shape starts to be a little bit oval. To end the comparison, the quantity of particles disturbed is lower for this figure than that of Figure 23a,b.

Finally, it needs to be kept in mind that although the temporal variation of circulation has been introduced based on experimental data, Figure 23 may not still represent the real behavior of the vortex, due to the absence of viscosity in the mathematical model employed, which would attenuate the flow particles' movement.

Some important details that need to be highlighted from Figure 23 are: At the time step of $t^* = 0.5$ (first row), it is seen how, as the circulation raises, the dimension of the vortices' central core increases. Additionally, the area of the perturbed flow field around the central core grows with the circulation, which is logical then the velocity induced is higher.

After $t^* = 1$ (second row), it is observed that the vortex of the medium airplane still maintains its axial position, while the heavy airplane vortices have slightly displaced towards the right. This is because of the horizontal velocity associated with the medium airplane, which is slower at the initial time steps than the one associated with the heavy ones; Figure 12 clarifies this point.

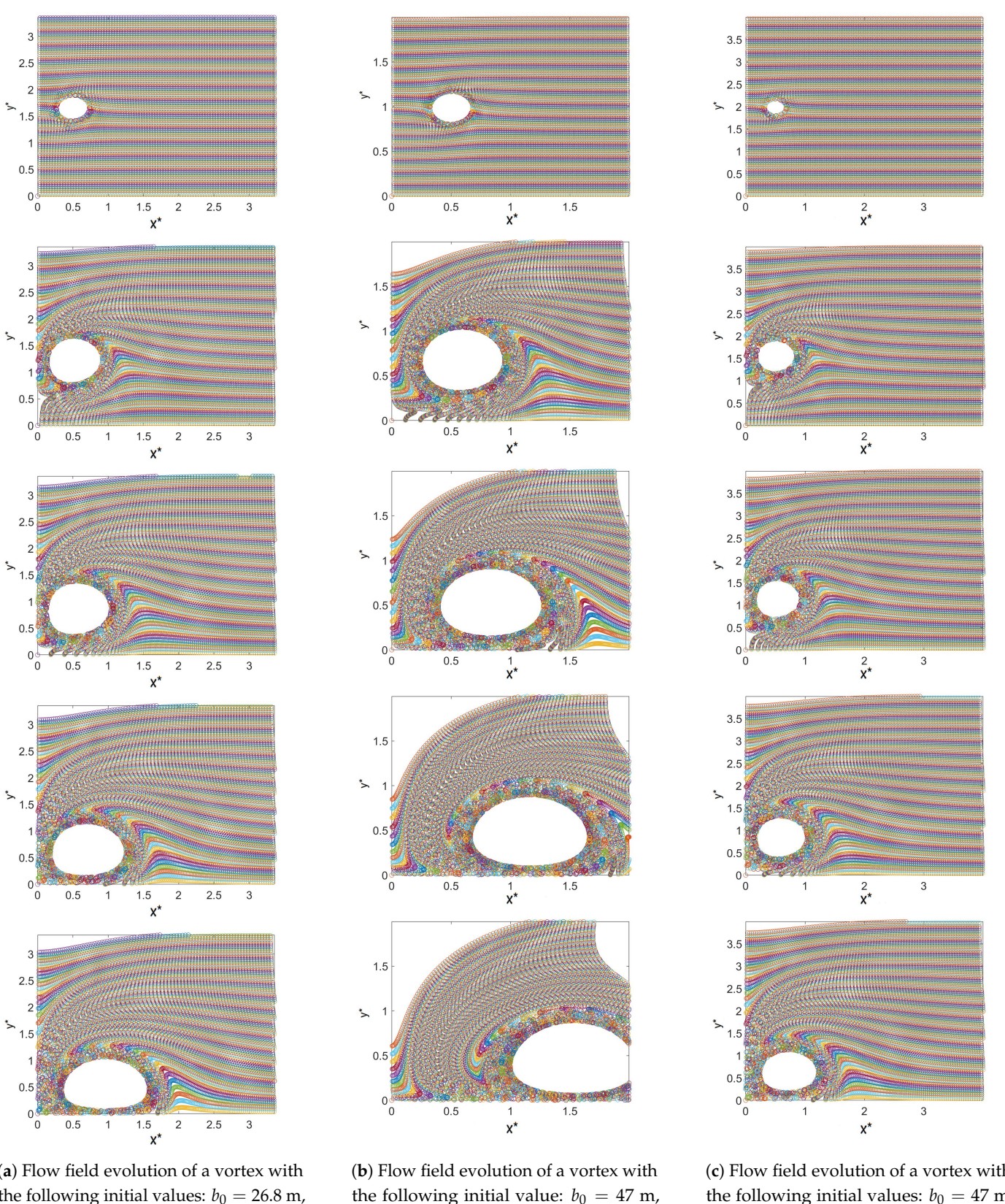

(**a**) Flow field evolution of a vortex with the following initial values: $b_0 = 26.8$ m, $\Gamma_0 = 250$ m$^2$/s, $y_0 = 45$ m.

(**b**) Flow field evolution of a vortex with the following initial value: $b_0 = 47$ m, $\Gamma_0 = 458$ m$^2$/s, $y_0 = b_0$.

(**c**) Flow field evolution of a vortex with the following initial values: $b_0 = 47$ m, $\Gamma_0 = 458$ m$^2$/s, $y_0 = 2b_0$.

**Figure 23.** The flow field representation has been performed for 5 time steps, which correspond (from top to bottom) to $t^* = 0$, 0.5, 1, 1.5 and 2. The ambient conditions include no crosswind.

At $t^* = 1.5$ (third row), the vortex of the medium airplane has a higher vertical speed (see also Figure 11); this is due to the fact that the two vortices are closer than for larger airplanes, and therefore the mutual influence is stronger (although the highest vertical velocity is the one of the vortex generated at a higher altitude). Additionally, the shape of its central core starts to deform, a fact that can be more clearly visualized at $t^* = 2$. The deformation of the flow field around the central core is now clear, becoming much more relevant at high circulations and as time increases. At this time step, it is also seen that once the vortices are near the ground, the horizontal central core displacement sharply rises, and the vortices quickly move away from each other. Based on Figure 10, it seems the central core's horizontal displacement speed increases as the vortex gets closer to the ground; in fact after about 60 s the vertical velocity becomes almost zero and the vortices' movement is fully controlled by the horizontal one. This is the case of Figure 23a,b, and in Figure 23c the vertical movement will last until approximately 125 s after its generation. This fact can be seen in Figures 11 and 12. The vortices' central core and surrounding area's deformation due to the ground effect can be clearly seen at this time.

### 4.6. Flow Field Evolution under Crosswind Conditions

The temporal flow evolution of the vortices' central core and surrounding area for different crosswind speeds and circulations is presented in Figure 24. Under crosswind conditions the entire flow-field also undergoes modifications. In order to see how it varies, the cases with a two-phase circulation temporary decay will be used in Equations (33), (34) and (36). At the initial time steps the flow evolution with and without crosswind is very similar; compare Figures 23 and 24. Once the time reaches $t^* = 0.5$ the differences start to be seen. At this time step, as the crosswind velocity increases, the perturbed flow field moves further to the right, and the area below the central core is particularly affected by the crosswind velocity where it generates a perturbed flow wedge that in the future times tends to wrap up the vortex central core. The central core flow field engulfment appears to be less relevant as the circulation increases; this is due to the higher flow field intensity associated with higher circulations.

At the time step of $t^* = 1$, the central core wrapping up is clearly observed; the vortex's central core starts suffering a clear deformation and elongation due to the stresses generated by the crosswind velocity. The deformation increases with the circulation decrease (taking into account that the deformation of the vortex generated by the medium airplane is similar to the other ones, but the time elapsed is lower, because $t_0$ is lower, see Table 5).

For the last two time steps ($t^* = 1.5$ and $t^* = 2$), the vortices under a crosswind velocity of 2 m/s present a very noticeable deformation, stretching the vortices and further elongating them, and the flow area upstream is completely swept towards the downstream direction. For the three different cases presented it is clearly seen how the crosswind deeply deforms the center of the vortex, passing from a round shape to an elliptical one. This mentioned degeneration is bigger as the time increases, and the deformation due to the ground effect can hardly be seen, and thus it loses relevance. What is particularly relevant to mention at the final time step presented is that the vortical structures have moved sideways (along the abscissa axis). The medium airplane vortices have moved, approximately, 110 m, and the ones of the heavy airplanes have displaced by almost 200 and 190 m for the case of the vortices generated at a higher altitude. The value for the same models but without considering crosswind is, approximately, 30, 75 and 40 m, respectively.

Another aspect to be observed is the tendency of the particles to move together with the crosswind, instead of rotating together with the vortex, a characteristic that differs from the observations made for the case without crosswind (Figure 23). Additionally, in all of the cases the particles of the right side of the represented domain move further sideways to the right than the ones in the left side. The same happens when comparing the top to the bottom of the domain.

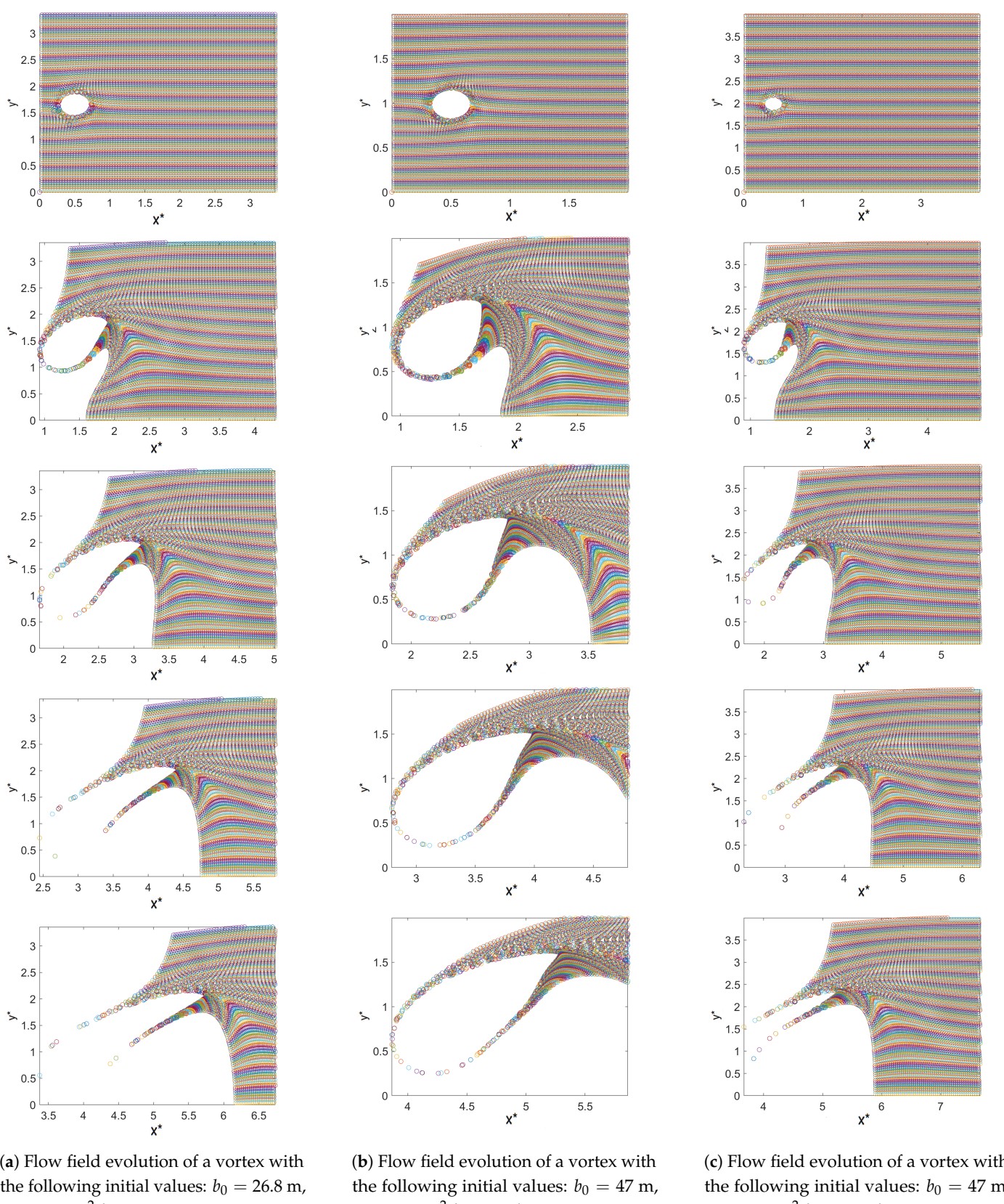

(**a**) Flow field evolution of a vortex with the following initial values: $b_0 = 26.8$ m, $\Gamma_0 = 250$ m$^2$/s, $y_0 = 45$ m.

(**b**) Flow field evolution of a vortex with the following initial values: $b_0 = 47$ m, $\Gamma_0 = 458$ m$^2$/s, $y_0 = b_0$.

(**c**) Flow field evolution of a vortex with the following initial values: $b_0 = 47$ m, $\Gamma_0 = 458$ m$^2$/s, $y_0 = 2b_0$

**Figure 24.** The flow field representation has been performed for 5 time steps, which correspond (from top to bottom) to $t^* = 0$, 0.5, 1, 1.5 and 2. The ambient conditions include a crosswind equal to 2 m/s.

One characteristic that has to be taken into account is that the boundary layer has not been represented in the near-ground domain, but in the real case could have some relevance for the crosswind movement, because the sideway movements of the particles touching the ground could generate new vortexes that could affect the motion of the principal ones.

Finally, the main difference between the cases with and without crosswind resides in noticing that the ground effect, which was particularly relevant in the cases without crosswind, becomes quite irrelevant as the crosswind velocity increases.

### 4.7. Time Reduction Analysis

In the actual regulations, the time between two consecutive airplanes is established by the ICAO. Both at take-off (Table 7) and landing (Table 8), see reference [19], this time is a bit conservative, because it is based on a norm that has to be followed independently of the conditions under which the procedures are being designed. The aim of this section is to evaluate how much time could be saved if instead of generalizing and including all of the different conditions in the same rule, each case is studied independently with the conditions of each airport.

**Table 7.** Time between departing airplanes, information extracted from reference [19].

| Following Airplane | Preceding Airplane | Departing Separation Minima |
|:---:|:---:|:---:|
| Light | Medium | 2 min |
| Medium | Heavy | 2 min |
| Light | Heavy | 2 min |
| Medium | A380 | 3 min |
| Light | A380 | 3 min |

**Table 8.** Time between arriving airplanes, information extracted from reference [19].

| Following Airplane | Preceding Airplane | Arriving Separation Minima |
|:---:|:---:|:---:|
| Light | Medium | 3 min |
| Medium | Heavy | 2 min |
| Light | Heavy | 3 min |
| Medium | A380 | 3 min |
| Light | A380 | 4 min |

In order to determine when the vortices leave the runway domain, Figure 25 was created, which for the different cases evaluated, shows the central core position along the abscissa axis over time. The equations characterizing the measured circulation decay were employed to generate this figure. The meaning of the legend presented in this figure is explained in Table 9. In this figure the runway width is presented as two horizontal lines. The times extracted from this graphic will be compared with the ones of reference [19], with the aim to see how much time can be reduced. The first thing to be observed is that the consideration of crosswind does not mean that the vortices will leave the runway earlier. Without crosswind, the vortices get away from the domain within a similar time to that when a crosswind velocity of 2 m/s is considered. On the other hand, the worst case is when the crosswind velocity is of 1 m/s; then for the A320 airplane (whose categorization is medium airplane), the vortices will leave the runway nearly 140 s after their formation. In the other case, for the heavy airplane (A340-300), the vortices leave the runway long after 180 s.

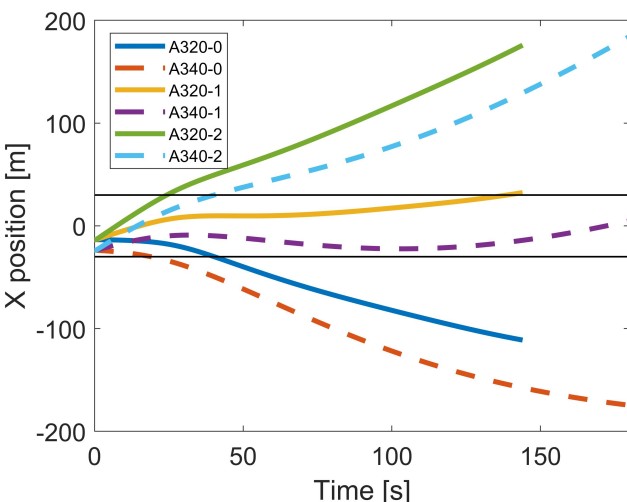

**Figure 25.** X position of the starboard vortex throughout time. Two models and three different crosswind velocities have been considered. The black lines represent the point at which the vortices will leave the runway.

Continuing with Figure 25, the next step is to extract the values from it. To round up the times, thus maintaining a small margin of error, an additional 15 s are added to the values retrieved from the graphic. As the widest runway is 60 m, it will be considered that the vortices leave it when the curve crosses the black line situated at 30 m. The results are presented in Table 9, in which it is seen that as the crosswind speed increases, the vortices of the medium airplane will leave the runway before those of the heavy one.

The temporal variation of $\Gamma$ in the case of the heavy airplane is the one stated in Equation (33). For the medium airplane, the temporal variation corresponds to the one shown in Equation (36). In both cases, as experimental data show (Reference [11]), the temporal variation of $\Gamma$ is not being affected by the crosswind. These temporal variations are the ones during the landing procedure.

**Table 9.** Calculated time for the vortex to leave the runway. Additionally, this time is presented in nondimensional form, the time elapsed during the calculation and the $\Delta x$ and $\Delta y$ used to perform the domain's discretization are included. In all cases a time step of 0.5 s was used.

| Type of Airplane | Time (s) | Time/$t_0$ | Program's Execution Time (s) | $\Delta x = \Delta y$ (m) |
|---|---|---|---|---|
| A320 - 0 ($U_\infty = 0$ m/s) | 35 | 1.9 | 19 | 0.1 |
| A340 - 0 ($U_\infty = 0$ m/s) | 55 | 1.8 | 17 | 0.1 |
| A320 - 1 ($U_\infty = 1$ m/s) | 150 | 8.3 | 18 | 0.1 |
| A340 - 1 ($U_\infty = 1$ m/s) | - | - | 19 | 0.1 |
| A320 - 2 ($U_\infty = 2$ m/s) | 40 | 2.2 | 19 | 0.1 |
| A340 - 2 ($U_\infty = 2$ m/s) | 55 | 1.8 | 18 | 0.1 |

In order to see which changes are produced by the introduction of the in-house program, the estimated time gain in percentage was calculated using Equation (38), which compares the times obtained by the in-house program with the ones defined in reference [19]. The results are presented in Table 10, and show how much time could be saved if the present in-house program is implemented at the airports. In most of the cases studied the time reduction oscillates between 50% and 80%. The increase in time for the cases with $U_\infty = 1$ m/s is due to the strength of the crosswind not being high enough to move away the port vortex from the runway in a quick manner. For this case, the vortex speed is contrary to the crosswind velocity and the difference in velocity between them is small. As a result, the vortex needs to cross the whole runway moving slowly.

$$\text{Time Gain}(\%) = \frac{\text{Time (in-house code)} - \text{Time (tables)}}{\text{Time (tables)}} \cdot 100 \tag{38}$$

**Table 10.** Calculated time gain under landing procedures and for different ambient crosswind conditions.

| Following Airplane | Preceding Airplane | Procedure | Estimated Time Gain |
|---|---|---|---|
| Light | Medium | Landing; $U_\infty = 0 \text{ m/s}$ | −80.56% |
| Light | Medium | Landing; $U_\infty = 1 \text{ m/s}$ | −16.67% |
| Light | Medium | Landing; $U_\infty = 2 \text{ m/s}$ | −77.77% |
| Light | Heavy | Landing; $U_\infty = 0 \text{ m/s}$ | −69.44% |
| Light | Heavy | Landing; $U_\infty = 1 \text{ m/s}$ | −% |
| Light | Heavy | Landing; $U_\infty = 2 \text{ m/s}$ | −69.44% |
| Medium | Heavy | Landing; $U_\infty = 0 \text{ m/s}$ | −54.17% |
| Medium | Heavy | Landing; $U_\infty = 1 \text{ m/s}$ | −% |
| Medium | Heavy | Landing; $U_\infty = 2 \text{ m/s}$ | −54.17% |

Finally, except for the case of $U_\infty = 1 \text{ m/s}$, the results should be taken as valid in the practical field, because the time elapsed does not widely exceed $t^* = 2$, and thus this stays inside the domain at which the in-house program is accurate, meaning that this application could be taken as reliable.

At this point it is interesting to highlight that when considering the methodology proposed by [3], consistent in reducing the effect of fluid viscosity via sucking the boundary layer, or via causing the main vortex to split in many other small ones by putting horizontal plates in the runway (Holzäpfel et al. [4]), the potential flow tool presented in this paper would be particularly useful, especially after considering its precision during the initial time steps, just before the vortex rebound starts. The tool presented in this paper needs to be seen as fast, widely applicable and accurate enough to estimate instantaneously the minimum time needed between two consecutive airplanes. The vortex behavior could be modeled even better if the secondary vortices' evolution was also included in the in-house code. This would allow to estimate the rebound effect and greatly improve the temporary results of the in-house program. It has been stated that the in-house code generates good results provided that the experimental circulation temporal decay is known. To extensively use the in-house code, the circulation temporal decay measured from different airplanes in different airports and under different ambient conditions needs to be known. A possible methodology, among others, to avoid performing such a large amount of measurements, would be using an Artificial Neural Network (ANN), and then the circulation decay for conditions not measured could be extrapolated. If this information could be implemented in the presented in-house program, the minimum time between any two consecutive airplanes under any atmospheric conditions and for any airport could be determined.

## 5. Conclusions

The method presented in this article provides a large amount of information regarding vortices' behavior and their temporal evolution. The lateral motion of the vortices and the velocity field both in the vertical and in the horizontal directions as well as the pressure field are clearly defined. The movement of the fluid particles has been introduced at several time steps, which was shown to be a very useful tool as can be seen when the perturbation on the domain starts to decay; in addition, it shows how the vortex's central core deforms over time.

The generated in-house code also considers the effect of the crosswind, showing that for crosswind velocities higher than 2 m/s, the ground effect becomes almost irrelevant and the entire flow field temporal evolution is dominated by the crosswind effect. The code generated is therefore applicable to a wide range of situations and can be particularized to any airport.

As seen in Section 4.7, the in-house program showed that a considerable reduction of time between consecutive airplanes is possible (see Table 10), therefore demonstrating a possible increase of airports' throughput. The estimated time gains need to be seen as very promising and applicable.

Another advantage of the generated in-house code is that it generates the results in less than one minute. Therefore, given ambient conditions and the type of airplane, the Air Traffic Control (ATC) could predict, using the present in-house code, the vortex behavior with sufficient anticipation to determine the lowest safe time between the leading aircraft and the following one.

**Author Contributions:** Conceptualization, J.M.B.; methodology, J.M.B. and J.M.D.; software, J.M.D.; validation, J.M.D.; formal analysis, J.M.D. and J.M.B.; investigation, J.M.D. and J.M.B.; resources, J.M.D.; data curation, J.M.D.; writing—original draft preparation, J.M.D.; writing—review and editing, J.M.B.; visualization, J.M.D.; supervision, J.M.B.; project administration, J.M.B. All authors have read and agreed to the published version of the manuscript.

**Funding:** This research received no external funding.

**Institutional Review Board Statement:** Not applicable.

**Informed Consent Statement:** Not applicable.

**Data Availability Statement:** Not applicable.

**Conflicts of Interest:** The authors declare no conflict of interest.

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
