# Peer review of "Airplane Vortices Evolution Near Ground"

_applsci, doi:10.3390/app11010457_

Round 1

Reviewer 1 Report

This is the review of a resubmitted paper which deals with motion of wing vortices near the ground. The authors employed potential flow theory and the effect of viscosity was implicitly taken into account by the temporal decay of circulation, which is obtained empirically as either a single-parameter exponential decay or 6th order polynomial fit. In this version included is the comparison against experimental data, which shows that once vortices approach the ground potential theory does not do a good job, as authors allude to in the text.  In view of this important limitation I am sceptical whether this work is worthy of publication. On the other hand, the approach is clear, it is described in detail and most results are meaningful and potentially useful. Therefore, I do not have strong objections against publication. However, having now read the paper several times I think it is too long for its purpose and that the practical impact of this work is overemphasized. Therefore I recommend further revision before acceptance if the editorial/review procedure continues.   

*** Major points ***

The abstract is too long and reads like a strained attempt to highlight the importance of this work for practical applications. However, the abstract does not correctly convey the limitations in the modelling; therefore it should be shortened and toned down.

I disagree with authors that experimental-based temporal vortex decay improves the accuracy of their program; results are very close together. I think this is an interesting finding of the present study given the simplicity of using the single-parameter model for calculations.

Section 2.1 is not really needed.

Figure 3: What is P2P in figure 3? Why are there 2 black solid lines in (b)? The dimensions y and z are inconsistent with the paper.

Velocity distributions in figures 6 and 23 are not really useful; could be omitted or juxtaposed with vectors.

Also particle motions in figure 25 and 26 are not meaningful and should be omitted given that the paper is too long.

The conclusions are not essentially evidence-based arguments but a discussion of how results could be implemented in a practical scenario. Therefore, I strongly suggest to merge this discussion with section 4.7 in a separate section (Discussions).

*** Minor points ***

Line 27: otherwise the risk of having a crash raise[s].

Line 38: … aeroplanes need[s] to be large enough …

Line 133: … multiplied by their conjugate … (delete d)

In equations the atan function should not be italicized.

Throughout the text “temporary variation” should be replaced by “temporal variation”.

Unnumbered line: “in-house program model” delete the word ‘model’.

Author Response

Please see the document attached.

Reviewer 2 Report

I course of time the paper has undergone large modifications which have eliminated the most important mistakes inside the paper and have increased the level of the paper. I recommend it for publication.

Author Response

Please see the document attached.

This manuscript is a resubmission of an earlier submission. The following is a list of the peer review reports and author responses from that submission.

Round 1

Reviewer 1 Report

The paper describes a study of wake vortices created from aircraft wings near ground. The approach is to assume the flow potential of two symmetric vortices near the ground and - from that potential - to obtain the velocity and pressure fields. The effect of different aircraft sizes and crosswind speeds are considered. The approach is over-simplistic since the effect of viscosity is  taken into account by introducing an arbitrary time scale for the decay in vortex circulation. However, the decay of vortex circulation depends on the parameters of the problem, which is the main difficulty in reality. The fallacy of the results can be seen in that the initial vortex circulation has no influence on the vortex trajectory, which does not comply with observations. Furthermore, the flow is treated as two dimensional whereas wing vortices are three-dimensional structures in reality. This means that neither the effect of their initial 3D structure nor winds in directions other than the crosswind can be taken into account in the analysis.  The authors conclude that “airports’ capacity could be increased noticeably if their approach could be finally applied in the real world” which clearly illustrates that they are unaware of the limitations of their approach. Generally, there are unclear points, missing information (e.g. the characteristic decay time), etc. The paper rather reads as an undergraduate project report/thesis, for which the student is to be commended. However, the manuscript is unsuitable for journal publication.  

Author Response

Please see the document attached.

Reviewer 2 Report

The authors proposed utilizing a combination of potential flow solver and empirical vortex decay model to estimate the wake-vortex strength evolution in ground vicinity. The empirical decay model was obtained from one of the references with few modifications (or at least the modifications were not sufficiently presented in the methodology section.) As the empirical equation and the data used for the validation is from the same original data set, I don't think it could be viewed as sufficiently validated. It would be better to show the validation against two-phase decay data from [14] for the same lead aircraft class. The authors did not demonstrate that the equation will hold true for the different Gamma_0 and b_0 values, especially in the case with high Gamma_0 and low b_0 where the viscous interaction between the vortices is important.

The authors also produced some analysis of varying characteristic values used in the equations, e.g. Gamma_0, b_0, but those same characters have been accounted for in previous experimental and simulation studies, where the different aircraft wake could be collapse into single line via normalizing the circulation by Gamma_0. I also believe that Sec. 4.4 could be greatly improved by comparison with the probabilistic model based on LIDAR measurements (e.g. https://doi.org/10.2514/1.C034287).

Regarding the time-saving analysis, the authors appears to have used a comparison between the established time-separation for given encounter pair and the time it takes for the vortex to clear the flight corridor. However, the author never established how good their vortex-movement prediction is when compared to empirical measurements or simulation data when the vortex is subjected to cross-wind, where the capturing or modeling of viscous interaction with the ground. The authors also did not include encounter-modeling in their analysis.

Overall, the research could be useful if properly validated and if the empirical model were able to properly capture the viscous interaction between the vortices and between the vortex and ground.

Author Response

Please see the document attached.

Reviewer 3 Report

The topic of this paper is modeling the vortices generated under aeroplane wings using the potential flow theory. The motivation is based on the assumption, that the CFD simulations require extremally high computational times to treat the same problem and their results depend on the turbulence model used. I do not think that modeling of a pair of vortices represents a real challenge for CFD, still the idea of the paper is not wrong. Also, the presented results are interesting from the application point of view. What is wrong in this paper is the theoretical background.

  • The viscosity effects are neglected. So, the dissipation of the vortices is described with an empirical formula, which must be calibrated from the experimental data. Also, the real vortex is simply described as the potential vortex with the rigid vortex core or with the constant velocity vortex core (it is not clear from the paper). In reality, near ground, more complex vortex structure is generated, which influences the temporal behavior of primary vortices.
  • In the mathematical background, there are some important mistakes. The key equations (22)-(23), which describe the velocity in the vortex center, include some formal mistakes. Also, these equations are derived from equations (19)-(20). But they cannot be derived from these equations without a thorough mathematical analysis. Figure 4 indicates, that authors used the empirical model of vortex with the constant velocity vortex core. But Figure 5 corresponds to the potential vortex with the rigid vortex core. Moreover, the tangential velocity (compared to the velocity components in the x- and y- directions) does not change the sign.   

Authors should give clearer mathematical background and provide better discussion concerning the numerical implementation (vortex model, time steps, computational times) and the calibration of the implemented empirical model in different situations and locations.

Author Response

Please see the document attached.

Round 2

Reviewer 1 Report

I have read the revised version of the paper and the authors’ responses to reviewer comments on the original version of the manuscript. The authors have refined their study and made substantial changes to the revised paper. They have also given reasonable answers to the reviewers’ critiques in the responses. I still believe that the model is over-simplistic to represent real  aeroplane vortices near the ground as remarked by all three reviewers. I am also not fully convinced by the validation tests reported in the revised version. Nonetheless, the paper describes well – despite some poor language – the methodology and the results, which could be of some practical interest. I tentatively recommend publication after minor revision to address the following issues:

Correct the term “vortice”; the singular form is “vortex”.

Revise the last two paragraphs in order to explain clearly how would “airport throughput could be increased”.  

Author Response

Please see the document attached.

Reviewer 2 Report

The authors has made significant changes to the manuscript and performed additional modeling, but failed to address the main issues from the first review, namely properly accounting for the viscous flow effect:

Since the potential flow code did not take into account of the viscous interaction between the vortex and the ground, the circulation decay and vortex trajectories are decoupled and simply follows whatever the empirical equations says. In this case, it's the rebound height that's been left out. It should be noted that the landing aircraft on ILS generally follows a pretty strict delta_h during landing, and the difference in rebound height for hovering vortex would definitely have an effect on the wake-encounter response of the follower aircraft. Since the separation time analysis never accounted for the height, luckily the data you get fits what you expected for time-saving.

I think the authors should consider the following before revising the paper: you're not using a non-dimentional/non-aircraft specific empirical equation for trajectory and circulation estimation, but instead utilize different empirical decay equation for different aircraft, making this approach not extendible to aircraft that you don't have data for. But, the position of the vortex is needed in order to measure circulation with the LIDAR setup, so in that case, would it not be easier to just use the statistical tracking data instead of the potential flow solver?

Author Response

Please see the document attached.

Reviewer 3 Report

The paper has undergone a large modification. A lot of new data and explanations are added.

Most of these changes are positive and have increased the level of the paper. Some minor grammar corrections of newly added texts are necessary.

But there still remained (already announced) formal mistakes inside the paper, which must be corrected:

  • The equations (31)-(32) /initially (22)-(23)/ still include formal mistakes (missing squares in denominator).
  • There is still missing better description of the vortex core treatment. Moreover, there is a formal mistake in Figure 7 /initially Figure 5/. It probably shows the shape of the velocity profile, or better, the graph of the velocity component Vy along the horizontal line located in the vortex center. This is not a graph of tangential velocity. Tangential velocity does not change the sign! Please add the graph of tangential velocity. Has the vortex the constant velocity core or the rigid vortex core? Specify explicitly.  

Author Response

Please see the document attached.
